# 2D MXenes polar catalysts for multi-renewable energy harvesting applications

Xiaoyang Pan [1] ✉, Xuhui Yang[2], Maoqing Yu[1,3], Xiaoxiao Lu[1,2], Hao Kang[2], Min-Quan Yang [2] ✉, Qingrong Qian[2], Xiaojing Zhao[1], Shijing Liang [3] ✉ & Zhenfeng Bian [4] ✉

The synchronous harvesting and conversion of multiple renewable energy sources for chemical fuel production and environmental remediation in a single system is a holy grail in sustainable energy technologies. However, it is challenging to develop advanced energy harvesters that satisfy different working mechanisms. Here, we theoretically and experimentally disclose the use of MXene materials as versatile catalysts for multi-energy utilization. $Ti_3C_2T_X$ MXene shows remarkable catalytic performance for organic pollutant decomposition and $H_2$ production. It outperforms most reported catalysts under the stimulation of light, thermal, and mechanical energy. Moreover, the synergistic effects of piezo-thermal and piezo-photothermal catalysis further improve the performance when using $Ti_3C_2T_X$. A mechanistic study reveals that hydroxyl and superoxide radicals are produced on the $Ti_3C_2T_X$ under diverse energy stimulation. Furthermore, similar multi-functionality is realized in $Ti_2CT_X$, $V_2CT_X$, and $Nb_2CT_X$ MXene materials. This work is anticipated to open a new avenue for multisource renewable energy harvesting using MXene materials.

The rapid depletion of fossil fuels and worsening environmental conditions have spurred continuous research endeavors in the conversion of intermittent solar, mechanical, and thermal energy sources into storable chemical energy[1–6]. Diverse renewable energy harvesting technologies, including piezo, photo, and thermal catalysis, have been developed over the past decades and have shown great potential for wastewater treatment and clean fuel generation[4–16]. Unfortunately, due to the unpredictable availability of single-source renewable energy, which depends on the season, climate, and geographical position[17,18], the conversion efficiency and stability provided by individual energy harvesting technologies are insufficient for practical deployment. Conventionally-designed energy harvesters normally can utilize only one energy source and are poorly effective for capturing other energy resources. For instance, $TiO_2$ has been widely investigated in

photocatalysis, but it shows no activity for piezo or thermal catalysis. Consequently, significant amounts of other forms of harvestable energy are wasted, thus hindering the maximization of the energy conversion capability.

To break through limitations, a promising strategy is to develop advanced energy harvesters that can capture multiple energy sources. Generally, hybrid devices are constructed by embedding individual harvesters made of different materials into one system, but this is restricted by various material incompatibility and fabrication challenges. As such, research efforts have departed from the traditional paradigm of fabricating hybrid devices and shifted to the development of single-composition energy harvesters. Nevertheless, realizing such multi-functionality remains daunting due to the need to simultaneously satisfy different energy conversion mechanisms within a single

[1]College of Chemical Engneering and Materials, Quanzhou Normal University, Quanzhou 362000, P. R. China. [2]College of Environmental and Resource Sciences, College of Carbon Neutral Modern Industry, Fujian Key Laboratory of Pollution Control & Resource Reuse, Fujian Normal University, Fuzhou 350007, P. R. China. [3]National Engineering Research Center of Chemical Fertilizer Catalyst Fuzhou University, Fuzhou 350002, P. R. China. [4]Education Ministry Key and International Joint Lab of Resource Chemistry and Shanghai Key Lab of Rare Earth Functional Materials, Shanghai Normal University, Shanghai 200234, China. ✉e-mail: xypan@qztc.edu.cn; yangmq@fjnu.edu.cn; sjliang2012@fzu.edu.cn; bianzhenfeng@shnu.edu.cn

material. Especially, different energy conversion effects should be independent or coupled, instead of counteracting each other.

In the last decade, MXenes, with a formula of $M_{n+1}X_nT_x$ (X is C or N, and $T_x$ are surface O, OH, and/or F groups), have expanded rapidly into an extensive family of two-dimensional (2D) materials[19–25]. The versatile compositions and rich chemistry of MXenes give them great potential for diverse energy storage applications. Specifically, the abundant surface groups of F, O, and OH reduce the symmetry of MXenes, which may create polar domains that allow the material to be activated by mechanical vibrations[26–29]. Additionally, MXenes with excellent metallic conductivity feature electronic properties similar to noble metals and can be used as thermal catalysts[30–33]. Furthermore, MXene exhibits intense surface plasmon excitation, ensuring efficient and broad solar light absorption[34–37]. These combined unique polar, electronic, and optical properties make MXene materials promising candidates for the simultaneous conversion of multisource mechanical, thermal, and solar energies. However, there is no report on the exploration of MXenes as a versatile catalyst for such a purpose.

Herein, we theoretically and experimentally validate the utilization of multiple energy sources by MXene materials. As a typical example, 2D $Ti_3C_2T_x$ MXene prepared by the HF etching of $Ti_3AlC_2$ showed efficient piezo, thermal, and photothermal catalytic activity under stimulation of diverse energy sources, including vibration, flow, heating, and NIR light. The material outperformed most reported catalysts, representing an outstanding material for diverse energy utilization. Superoxide and hydroxyl species were generated during these processes. The synergistic effects of piezo-thermal and piezo-photothermal catalysis further improved the performance of $Ti_3C_2T_x$. Similar to $Ti_3C_2T_x$, the $Ti_2CT_x$, $V_2CT_x$, and $Nb_2CT_x$ MXene materials showed promising multifunctional catalytic properties.

## Results
### Theoretical and experimental study of MXene

To explore the possibility of using MXenes as multisource energy harvesters, a series of density functional theory (DFT) calculations and experimental characterizations were carried out to investigate their polar, electronic, and optical properties. $Ti_3C_2T_x$ are the most common MXenes, so they were studied first. As shown in Fig. 1a–d, typical $Ti_3C_2T_x$ structure models terminated without or with different OH, F, and mixed F-OH functional groups were constructed. Generally, piezoelectricity originates from the non-centrosymmetric nature of a material, which leads to the generation of electric dipoles. Therefore, the electric dipole moment is a key indicator of piezoelectric materials. To reveal the possible piezoelectric properties of $Ti_3C_2T_x$, the dipole moments of these structures were calculated. Pure $Ti_3C_2$ without functional groups showed neither in-plane nor out-of-plane dipole moments due to its centrosymmetric structure (Fig. 1a). When $Ti_3C_2$ was functionalized with only a single functional group, the resultant $Ti_3C_2T_x$ possessed a 2D hexagonal crystal structure composed of a Ti-C skeleton and surface terminal T (T=OH or F) groups. If the functional groups were located at a symmetrical position of the $Ti_3C_2T_x$, there would be no dipole moment. However, if the functional group was introduced at an asymmetric position, a permanent dipole moment perpendicular to the 2D molecular plane was observed (values of −0.088 and 0.038 eÅ were observed for F- and OH-terminated $Ti_3C_2$, respectively), indicating obvious piezoelectricity (Fig. 1b, c and Supplementary Table 1). Moreover, when two or more kinds of functional groups were present, regardless of whether they were located in symmetrical (Fig. 1d) or asymmetrical sites (Supplementary Fig. 1), an obvious permanent dipole moment was induced (Supplementary Table 1). We have also investigated the influence of the number of layers on the dipole moment of $Ti_3C_2T_x$. The dipole moment of multilayer $Ti_3C_2T_x$ (5 layers) was comparable to that of monolayer $Ti_3C_2T_x$,

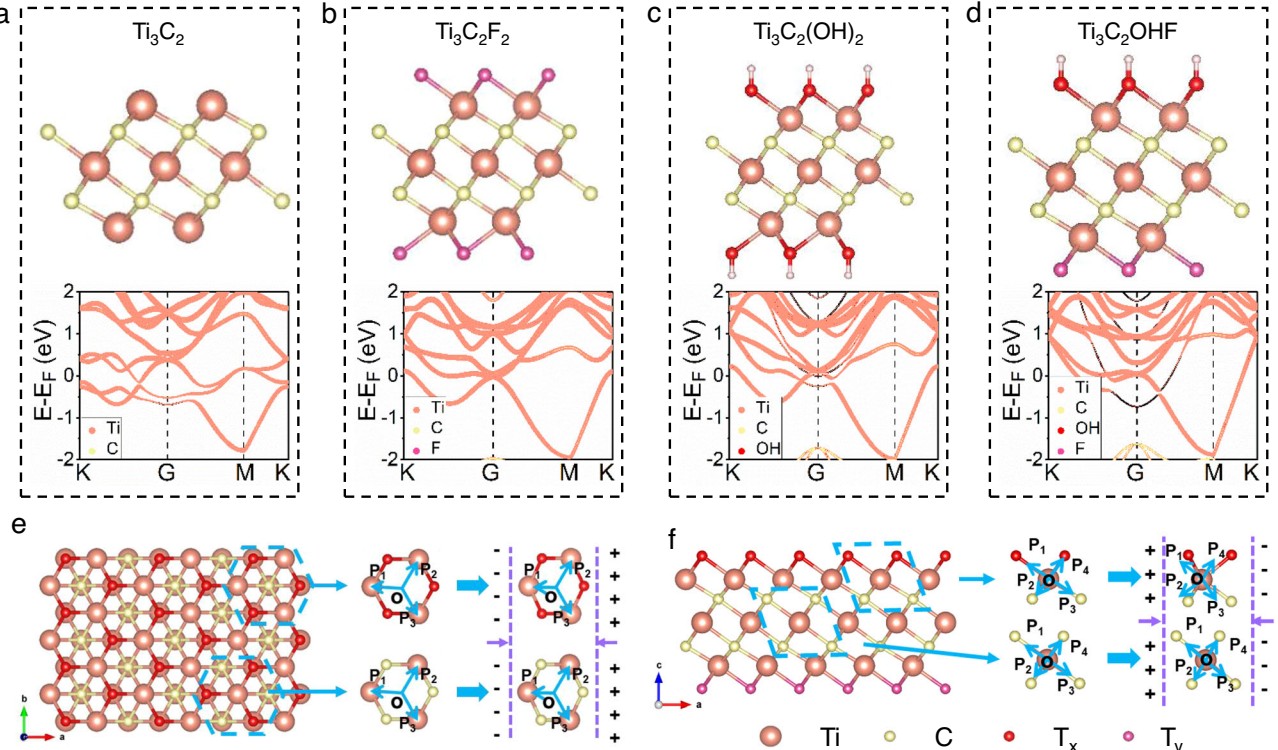

**Fig. 1 | Energy bands and crystal structures of $Ti_3C_2$ and $Ti_3C_2T_x$. a–d** Calculated energy bands and dipole moments of $Ti_3C_2$ and $Ti_3C_2T_x$ monolayers. **e** Top view of the monolayer $Ti_3C_2T_x$ crystal structure with the tensile deformation of the simplified $T_x$-Ti hexagonal structure (upper) and Ti-C hexagonal structure (lower), causing the piezoelectricity. **f** Side view of the monolayer $Ti_3C_2T_x$ crystal structure and the simplified model of the crystal structure deformation during a tensile process in the x-z plane of $T_x$-Ti (upper) and Ti-C (lower) bonds, causing the piezoelectricity.

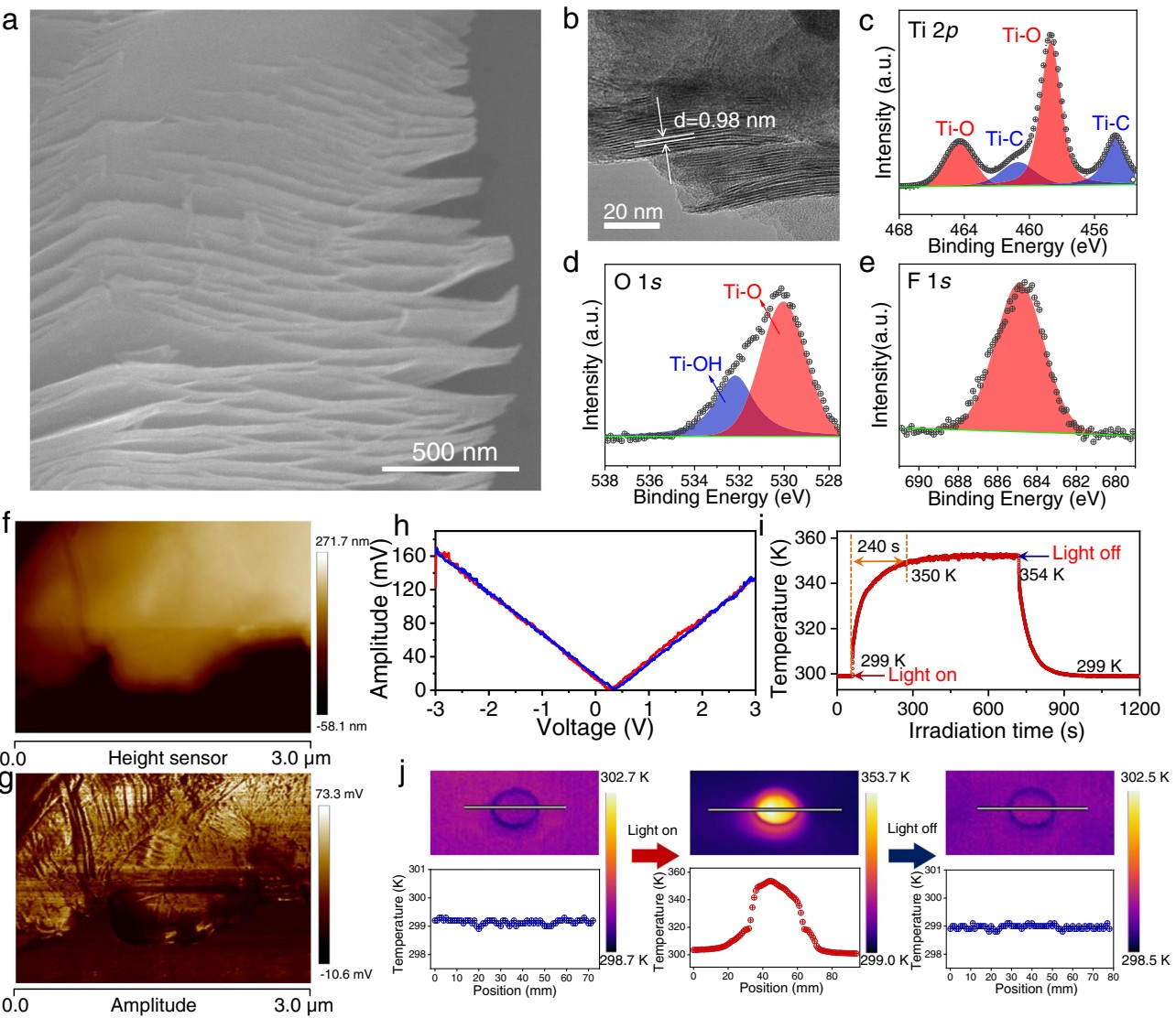

**Fig. 2 | Characterization of Ti₃C₂Tₓ. a** SEM image, **b** HRTEM image, XPS spectra of Ti₃C₂Tₓ: **c** Ti *2p*, **d** O *1s*, **e** F *1s*. **f** AFM image, **g** PFM image of Ti₃C₂Tₓ. **h** Amplitude-voltage curve of Ti₃C₂Tₓ. **i** Light irradiation-temperature curve of Ti₃C₂Tₓ. **j** The thermal images of the Ti₃C₂Tₓ.

suggesting that the number of layers had little effect on the dipole moment (Supplementary Table 2 and Supplementary Note 1).

The transformation from non-piezoelectric $Ti_3C_2$ to piezoelectric $Ti_3C_2T_X$ originated from changes in the crystal structure. Based on the structure models, monolayer $Ti_3C_2$ without functional group belongs to the P_3m1 space group, which exhibits an inverted symmetry and is intrinsically non-piezoelectric. As the surface was functionalized by $T_X$ groups (T=F, OH, O), the Wyckoff positions of the $Ti_3C_2T_X$ materials changed. The space group of the MXene changed from P_3m1 to P3m1, which is non-centrosymmetric, therefore giving $Ti_3C_2T_X$ piezoelectric properties (Supplementary Table 3). As specifically illustrated in Fig. 1e, from the *x-y* plane (top view), $Ti_3C_2T_X$ has a honeycomb lattice structure. From the side view, it is clear that the atomic structure can be divided into Ti-C combinations in the center and Ti-$T_X$ functional groups at both terminals (Fig. 1f). Because the Ti-$T_X$ functional groups were located at non-equivalent positions, the crystal structure obviously lacks the inverted symmetry center along the *x-y* plane, which gave $Ti_3C_2T_X$ MXene in-plane piezoelectric properties, i.e., the generation of a net polarization along the *x-y* plane[28]. This may lead to obvious piezoelectricity of the $Ti_3C_2T_X$ structure[27,28]. In addition, DFT calculations revealed that $Ti_3C_2T_X$ with different functional groups (F, OH) all exhibited metallic energy band structures (Fig. 1). The metallic

MXene demonstrated high thermal catalytic dehydrogenation performance comparable to that of noble metals[31].

Based on the theoretical calculations, we prepared the $Ti_3C_2T_X$ MXene by chemically etching $Ti_3AlC_2$ using an HF solution. Supplementary Fig. 2 shows the XRD patterns of the $Ti_3AlC_2$ precursor and $Ti_3C_2T_X$ product. After HF etching, the characteristic peaks of $Ti_3AlC_2$ from 33° to 43° disappeared. Compared with $Ti_3AlC_2$, the (002) peak was broadened and shifted towards a smaller angle. These changes confirmed that Al atoms were successfully extracted from $Ti_3AlC_2$[38]. Figure 2a, b shows the morphology and microstructure of the $Ti_3C_2T_X$ sample characterized by scanning electron microscopy (SEM) and transmission electron microscopy (TEM). $Ti_3C_2T_X$ possessed a stacked-layer structure with opened interspaces. The layer spacing was determined to be 0.98 nm, as shown in the HRTEM image (Fig. 2b). Moreover, the chemical composition and surface state of $Ti_3C_2T_X$ were investigated by X-ray photoelectron spectroscopy (XPS). As illustrated in Fig. 2c, two pairs of peaks are observed in the Ti *2p* spectrum, corresponding to the $2p_{1/2}$ and $2p_{3/2}$ orbitals of Ti(III) and Ti(IV) species[39]. The high-resolution C *1s* spectrum shows two main peaks at 281.6 and 284.6 eV, which were assigned to C-Ti and C-C bonds, respectively (Supplementary Fig. 3)[39]. Notably, the O *1s* spectrum (Fig. 2d) reveals two peaks, one at 529.9 eV belonging to O-Ti bonds

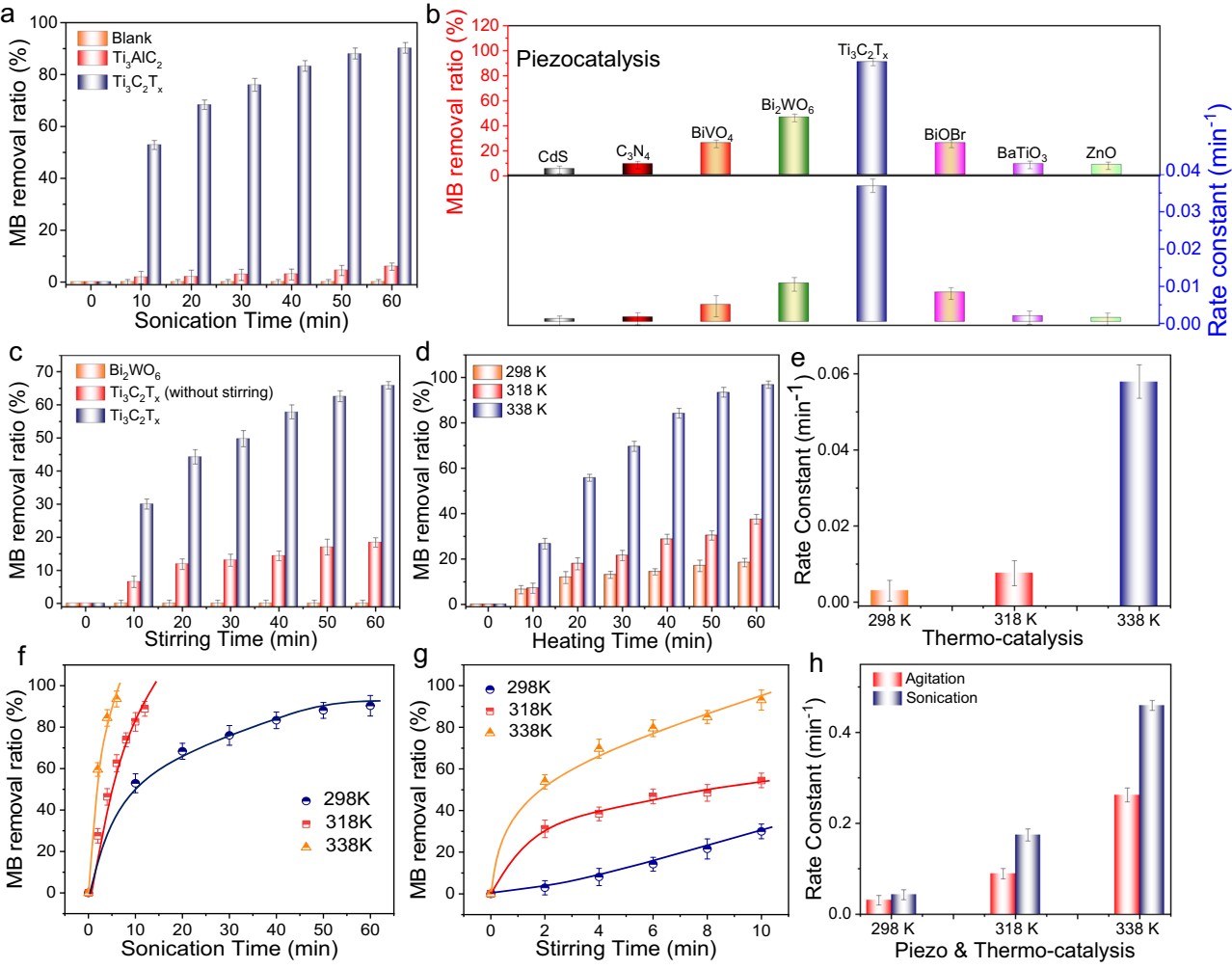

**Fig. 3 | Catalytic degradation of MB. a** The piezocatalytic efficiency of methylene blue (MB) degradation over $Ti_3AlC_2$ and $Ti_3C_2T_X$. **b** MB removal ratio and the corresponding rate constants over different polar semiconductors after 60 min sonication. **c** Piezocatalytic degradation of MB on different samples at 298 K under continuous stirring (1000 rpm) in the dark. **d**, **e** Thermal catalytic degradation of MB over $Ti_3C_2T_X$ and the corresponding rate constants. **f–h** Piezo-thermal catalytic degradation of MB over $Ti_3C_2T_X$ and the corresponding rate constants. All the data in (**a–h**) were collected three times, and the error bars represent the standard deviation.

and another at 532.2 eV, indexed to -OH bonds. In addition, $F^-$ ions were also detected on the surface of $Ti_3C_2T_X$ (Fig. 2e). The results suggest the successful preparation of MXene $Ti_3C_2T_X$, which was functionalized with F, O, and OH groups. In light of the theoretical calculations, the arrangement of the different functional groups broke the symmetry of the $Ti_3C_2T_X$, regardless of their location. Thus, a piezoelectric effect is expected in the HF-etched $Ti_3C_2T_X$.

To further experimentally investigate the piezoelectricity of the $Ti_3C_2T_X$, piezoresponse force microscopy (PFM) was carried out. $Ti_3C_2T_X$ material with a thickness of ca. 140 nm was clearly detected in the AFM image (Fig. 2f and Supplementary Fig. 4). The relative amplitude image (Fig. 2g) and typical butterfly amplitude loop (Fig. 2h) confirmed the piezoelectricity of $Ti_3C_2T_X$[40]. Based on the amplitude-voltage curve, the piezoelectric coefficient of $Ti_3C_2T_X$ was calculated to be 192.38 pm/V. Therefore, it could be utilized as a catalyst to capture mechanical energy.

The optical properties of $Ti_3C_2T_X$ samples were measured by diffuse reflectance spectroscopy (DRS). As presented in Supplementary Fig. 5a, $Ti_3C_2T_X$ showed efficient light absorption in the UV-Vis-NIR region with a notable surface plasmon resonance (SPR) peak, similar to that of noble metal nanostructures. Literature reports have shown that the plasmon energy is determined by an interplay between interband transitions and boundary effects correlated with the size and thickness

of $Ti_3C_2T_X$ flakes. This offers a potential method to tune the plasma frequencies over a large spectral range from the visible to near-infrared region[41–43]. Multiple plasmon resonance modes, including dipole and multipolar, were observed over a wide range of resonance wavelengths in $Ti_3C_2T_X$ flakes[43]. Moreover, the light reflectance of $Ti_3C_2T_X$ from 250 to 2000 nm was lower than 15% (Supplementary Fig. 5b), implying strong light absorbance (>85%) of the material within the solar spectrum. Because of the high light harvesting properties, the photothermal effect of $Ti_3C_2T_X$ was measured. As shown in Fig. 2i, j, under simulated 1 sun irradiation, the temperature of the $Ti_3C_2T_X$ rose rapidly from 299 to 350 K within 240 s and cooled down to 299 K when the simulated solar light was switched off, validating a significant light-induced heating effect. The broadband light capture ability, along with the excellent photothermal conversion, highlights the potential of using $Ti_3C_2T_X$ for solar energy harvesting and utilization.

## Piezo and thermal catalytic performance of MXene
Encouraged by the above theoretical and experimental investigations, the catalytic applications of the $Ti_3C_2T_X$ MXene sample was systematically measured under stimulation by diverse energy resources to evaluate its ability to use multiple energy sources. As shown in Fig. 3a, $Ti_3C_2T_X$ was first evaluated under sonication for methylene blue (MB) degradation to test its piezocatalytic performance. In the absence of

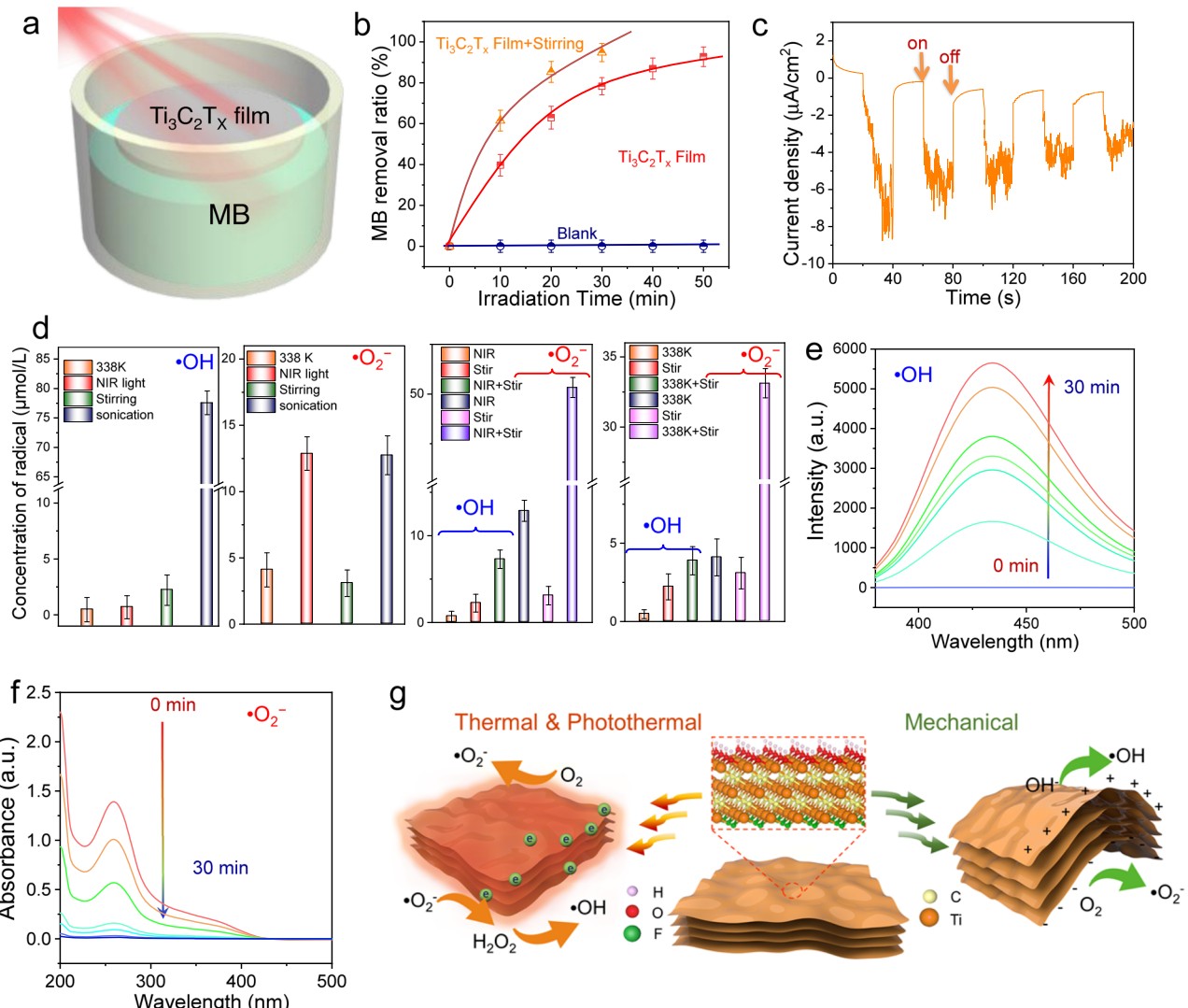

**Fig. 4 | Photothermal catalytic degradation of MB over Ti₃C₂Tₓ and reaction mechanism. a** Schematic diagram of the photothermal catalysis test. **b** Photothermal catalytic degradation of MB under NIR light irradiation of 700–1200 nm. **c** Transient current response of Ti₃C₂Tₓ under sonication. **d** The concentration of •OH and •O₂⁻ over Ti₃C₂Tₓ under different reaction conditions within 30 min. **e** Fluorescence spectra of TA solution over Ti₃C₂Tₓ under sonication at 338 K. **f** The absorbance of NBT molecule over Ti₃C₂Tₓ under sonication at 338 K. **g** Reaction mechanism of Ti₃C₂Tₓ for organics degradation under the stimulation of different energy sources. All the data in (**b**, **d**) were collected three times, and the error bars represent the standard deviation.

Ti₃C₂Tₓ, the absorption peak of MB was almost unchanged within 60 min, indicating that MB was difficult to decompose by sonication. However, with the assistance of Ti₃C₂Tₓ, MB was rapidly degraded upon increasing the reaction time and almost disappeared after sonication for 60 min (Supplementary Fig. 6). The MB removal efficiency reached 91%. No peak shift or new peaks occurred at low wavelengths, suggesting thorough degradation of MB. In addition, the IR spectra of the Ti₃C₂Tₓ before and after the catalytic reaction were measured to study the dye adsorption on the catalyst surface (Supplementary Fig. 7). No typical band for MB was observed. The identical IR spectra indicate that MB was degraded rather than adsorbed on the surface of Ti₃C₂Tₓ. The corresponding total organic carbon (TOC) content of the MB solution before and after the piezocatalytic reaction was measured to be 1.9 and 0.45 mg/L, respectively. The result further verified that most MB was degraded to $CO_2$, showing the excellent piezocatalytic performance of the Ti₃C₂Tₓ driven by mechanical energy. Moreover, in the presence of Ti₃AlC₂, the absorption peak intensity of MB was slightly decreased. The weak activity of Ti₃AlC₂ might be caused by the leaching of Al during sonication and surface functionalization by oxygen-containing functional groups. As evidenced by the ICP analysis

(Supplementary Note 2), noticeable $Al^{3+}$ was detected in the reaction solution of Ti₃AlC₂ after the catalytic activity test (0.5 mg/L). In addition, the XPS characterization of the used Ti₃AlC₂ showed that obvious Ti-O bonds were generated on the surface (Supplementary Fig. 8), indicating the partial transformation of Ti₃AlC₂ to Ti₃C₂Tₓ with O and OH groups during the catalytic reaction. This was the reason for the weak activity of Ti₃AlC₂.

To highlight the remarkable piezocatalytic performance of Ti₃C₂Tₓ, a variety of semiconductor piezocatalysts was prepared (Supplementary Figs. 9–16) and used to degrade MB under identical conditions. These catalysts are widely studied piezoelectric materials with considerable catalytic performance, according to a literature survey. As shown in Fig. 3b, Ti₃C₂Tₓ revealed much higher activity than reported piezoelectric semiconductors (Supplementary Fig. 17). Also, the catalytic performance was higher than most reported piezocatalysts in the literature (Supplementary Table 4). Taking the optimal polar Bi₂WO₆ nanosheet semiconductor as an example, it displayed the best piezocatalytic MB degradation with a rate constant of 0.012 min⁻¹ (Fig. 3b). Nevertheless, the MB degradation was only one-third of that obtained using Ti₃C₂Tₓ. In light of the high performance of

$Ti_3C_2T_X$ under sonication, its piezocatalytic activity was further evaluated under hydraulic forces driven by magnetic stirring (1000 rpm) because the flow is more common and gentler mechanical energy in natural environments[44]. The performance of the optimal $Bi_2WO_6$ semiconductor was also investigated as a comparison. As shown in Fig. 3c, $Bi_2WO_6$ exhibited almost no catalytic activity under stirring. In sharp contrast, $Ti_3C_2T_X$ removed nearly 70% of MB within 60 min with the assistance of stirring in the dark (Supplementary Fig. 18). These results verify that $Ti_3C_2T_X$ MXene is an excellent candidate for harvesting and converting mechanical energy.

During the piezocatalytic tests, we found that in the presence of $Ti_3C_2T_X$ without sonication or stirring, MB was still gradually degraded at 298 K in the dark (Fig. 3c). Because the surface area of $Ti_3C_2T_X$ was very low (2.8 $m^2$/g, Supplementary Table 5), and the catalytic reaction was carried out after pre-adsorption treatment, adsorption was unlikely to be the reason for MB removal. To confirm this inference, control experiments were performed. The surface charge of the $Ti_3C_2T_X$ was first investigated by zeta potential analysis (Supplementary Fig. 19a), which showed that the surface of MXene was negatively charged. Nevertheless, due to the small surface area of $Ti_3C_2T_X$, the adsorption capacities for three different dyes were low (Supplementary Fig. 19b). In addition, the possible influence of mass transfer on the adsorption process was also investigated (Supplementary Fig. 19c). The adsorption capacity of MXene without stirring was comparable to that of MXene during low-speed stirring (50 rpm). Therefore, it is suggested that $Ti_3C_2T_X$ can utilize ambient heat to catalyze MB degradation rather than adsorbing MB.

Moreover, the thermal catalytic degradation of MB at different temperatures was been investigated over $Ti_3C_2T_X$ MXene. As presented in Fig. 3d, MB was efficiently removed from 298 to 338 K. Increasing the temperature accelerated the reaction. At 338 K, MB was completely removed within 60 min without stirring, and the rate constant was determined to be 0.06 $min^{-1}$ (Supplementary Fig. 20 and Fig. 3e). In contrast, most polar semiconductors were completely inactive at 338 K (Supplementary Fig. 21). The efficient thermal catalytic activities might be attributed to the metallic property of $Ti_3C_2T_X$ MXene, which could activate molecular oxygen to generate reactive oxygen species. This was validated in the following mechanism study.

The bi-functionality of $Ti_3C_2T_X$ MXene inspired us to further investigate its piezo-thermal catalytic activity. As revealed in Fig. 3f–h, under sonication or constant stirring, the existence of heating further enhanced the performance of the $Ti_3C_2T_X$. The sample almost completely decomposed MB within 10 min at 338 K. The degradation efficiency of $Ti_3C_2T_X$ via the piezo-thermal effect was also much higher than the superposition of the piezocatalysis and thermal catalysis (Fig. 3h and Supplementary Fig. 22), indicating a synergistic effect between the two catalytic processes. Furthermore, $Ti_3C_2T_X$ was tested for the piezo-thermal catalytic degradation of methyl orange (MO) and rhodamine B (RhB). An analogous photoactivity enhancement was observed (Supplementary Fig. 23 and Supplementary Note 3). The results demonstrate the generality of the synergistic piezo-thermal catalytic effect of the $Ti_3C_2T_X$ for synchronous mechanical and thermal energy utilization.

### Photothermal and photothermal-piezocatalytic degradation of MB

$Ti_3C_2T_X$ displayed thermal catalytic performance, and DRS analysis of $Ti_3C_2T_X$ demonstrated broad solar light capture ability and excellent photothermal conversion. As such, it is anticipated that $Ti_3C_2T_X$ can be used for solar-driven photothermal catalysis. To facilitate the light-to-heat conversion, a floating $Ti_3C_2T_X$ film was prepared (Supplementary Fig. 24 and Fig. 4a), which showed highly efficient photothermal conversion by avoiding light loss caused by solution absorption. Moreover, the floating film localized thermal energy at the air/water interface, resulting in a rapid surface temperature increase. As

displayed in Supplementary Fig. 25, when irradiated by a commercial near-infrared (NIR) lamp for 20 min, the surface temperature of the $Ti_3C_2T_X$ film increased from 299 to 330 K, while the solution temperature only moderately increased to about 310 K. Correspondingly, a high photothermal catalytic activity for MB degradation was obtained (Fig. 4b). Within 50 min, MB was almost completely removed without stirring. In addition, the catalytic activity was further improved through constant stirring. The rate constant reached 0.097 $min^{-1}$ (Supplementary Fig. 26 and Supplementary Note 4), which was much higher than the values of the reported NIR-driven photocatalysts (Supplementary Table 6). Furthermore, a series of traditional piezocatalysts was prepared for comparison (Supplementary Fig. 27a). No NIR activity was detected for these catalysts because they were all incapable of harvesting NIR light (Supplementary Fig. 27b). A further comparison indicated that the degradation activity of $Ti_3C_2T_X$ was even higher than most of the visible-light-driven photocatalysts (Supplementary Fig. 28). These results show the excellent NIR light capture and utilization efficiency of $Ti_3C_2T_X$ MXenes.

To clarify the origin of the catalytic activity under NIR irradiation, the action spectrum of $Ti_3C_2T_X$ was obtained. As shown in Supplementary Figs. 29, 30a, $Ti_3C_2T_X$ displayed an action spectrum that resembled the plasmonic absorption spectrum of the sample. Importantly, the catalytic activity trends were in accordance with temperature changes caused by irradiation with different wavelengths (Supplementary Fig. 30b). Moreover, when controlling the reaction temperature at 298 K using a circulating cooling bath to eliminate the effect of light heating, the activity of $Ti_3C_2T_X$ under NIR light irradiation remained almost the same as its activity at 298 K without light irradiation (Supplementary Fig. 31). These results verify that the NIR light-induced activity of the $Ti_3C_2T_X$ MXene was caused by SPR photothermal conversion.

## Discussion

In theory, piezocatalysts will bend and deform when subjected to mechanical vibrations, which generate positive and negative charges due to the piezoelectric effect. The positive and negative charges polarize and produce a piezoelectric field in the corresponding direction that separates charge carriers to drive catalytic redox reactions. Thus, to reveal the underlying mechanism for the piezocatalytic activity of $Ti_3C_2T_X$, the piezo-current response of the sample was first measured to determine the generation of free charges under mechanical vibration. As shown in Fig. 4c, when ultrasonication was initiated on the $Ti_3C_2T_X$ electrode, a current was immediately generated. Once sonication was turned off, the piezoelectric current quickly terminated, confirming that the free charges were triggered by sonication. Since the catalytic process was carried out in the air in an aqueous solution, the piezo-generated charges transferred to the surface of $Ti_3C_2T_X$ and reacted with surface-absorbed species such as $O_2$ and $H_2O$ to produce reactive superoxide anions ($\cdot O_2^-$) and hydroxyl radicals ($\cdot OH$), as illustrated in Fig. 4d. This degraded organic pollutants. The charge transfer and reaction processes were similar to those of photocatalysis. Supplementary Fig. 32a shows the photoluminescence (PL) spectra of the $Ti_3C_2T_X$ dispersion tested in the presence of terephthalic acid (TA) probing molecules under sonication. The strong PL emission peak directly verified the generation of $\cdot OH$ radicals in the reaction system. Nitroblue tetrazolium (NBT) was used as a probe to quantify the $\cdot O_2^-$ concentration generated during the reaction. As shown in Supplementary Fig. 32b, an obvious decrease in the absorbance of NBT centered at 259 nm was observed, indicating that $Ti_3C_2T_X$ exhibited efficient activity for transforming $O_2$ into $\cdot O_2^-$. These results confirmed that $\cdot O_2^-$ and $\cdot OH$ radicals were generated by piezocatalysis, which were the main active species during dye degradation (Supplementary Fig. 33).

Similar PL and UV-vis absorption spectra were observed for the $Ti_3C_2T_X$-NBT and $Ti_3C_2T_X$-TA systems under heating (Supplementary

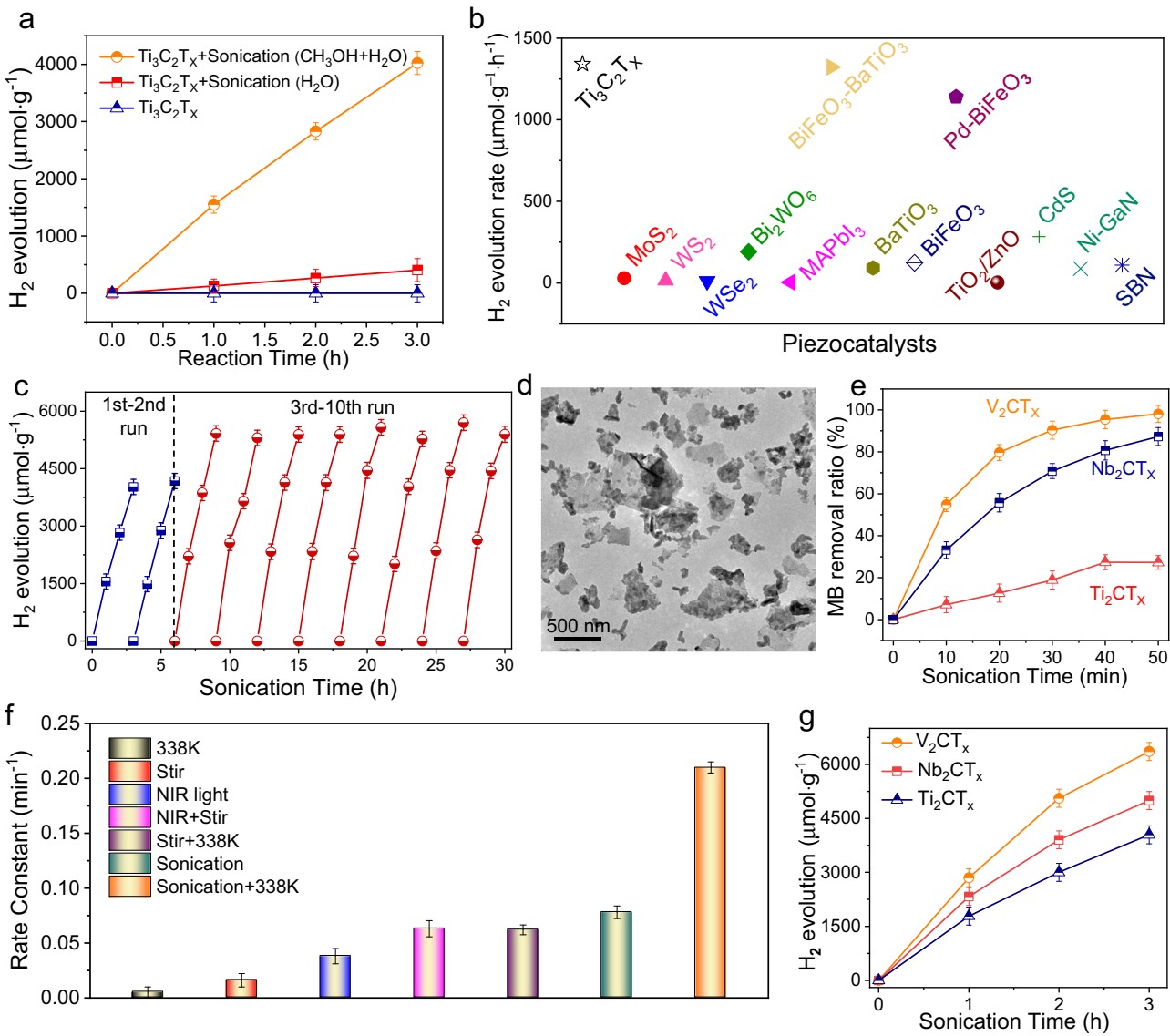

**Fig. 5 | Catalytic H$_2$ production and MB degradation over MXenes. a** H$_2$ production of Ti$_3$C$_2$T$_X$ under sonication in the dark. **b** Comparison of H$_2$ evolution performance of Ti$_3$C$_2$T$_X$ with some reported typical piezocatalysts. **c** The stability of Ti$_3$C$_2$T$_X$ under sonication. **d** TEM image of Ti$_3$C$_2$T$_X$ after ten cycles reaction. **e** Catalytic degradation of MB over the MXene materials under sonication conditions. **f** Catalytic degradation of MB over the V$_2$CT$_X$ under different reaction conditions. **g** H$_2$ production of V$_2$CT$_X$, Nb$_2$CT$_X$ and Ti$_2$CT$_X$ under sonication in the dark. All the data in (**a–c**, **e–g**) were collected for three times, and the error bars represent the standard deviation.

Fig. 34 and Supplementary Note 5), validating that thermal energy excited Ti$_3$C$_2$T$_X$ to generate charge carriers and form •O$_2^-$ and •OH radicals. The thermal catalytic activity was attributed to the metallic property of Ti$_3$C$_2$T$_X$. It could activate molecular oxygen to generate •O$_2^-$, which was further converted to •OH via the formation of an H$_2$O$_2$ intermediate (Supplementary Equations (1–3))[45]. To further understand the underlying mechanism of the thermal catalytic process, DFT calculations were carried out. Theoretically, the thermal activation of O$_2$ to form •O$_2^-$ can be investigated by calculating the Bader charge difference between free O$_2$ and O$_2$ adsorbed on the surface of a catalyst. Generally, the charge transfer of 0.5 |e| is enough to generate •O$_2^-$[46]. Supplementary Fig. 35 and Supplementary Table 7 show the DFT calculation results of changes in the Bader charge between the free O$_2$ molecule and that in monolayer and bilayer Ti$_3$C$_2$T$_X$. The charge density differences reveal that electrons may have transferred from Ti$_3$C$_2$T$_X$ to adsorbed O$_2$ molecules. The charge-transfer quantities were 0.85 and 0.83 |e| for monolayer and bilayer Ti$_3$C$_2$T$_X$, respectively, confirming that Ti$_3$C$_2$T$_X$ could thermally activate O$_2$. Notably, the

concentration of free radicals (•O$_2^-$ and •OH radicals) produced by Ti$_3$C$_2$T$_X$ under heating at 338 K was lower than that obtained by sonication at 298 K. The two catalytic processes showed similar activities for MB degradation because increasing the reaction temperature facilitated the production of oxygen radicals and also accelerated the reaction according to the Arrhenius formula: $k = Ae^{-E_a/RT}$ (where, $k$ is the rate constant, $E_a$ is the activation energy, and $T$ is the reaction temperature).

Active species •O$_2^-$ and •OH were also detected during the photothermal catalytic process (Supplementary Fig. 36), which induced the degradation of MB. The reaction mechanism was similar to the thermal catalytic process, for which solar energy was converted to heat through plasmonic MXene and then thermally activated MXene. In addition, the combination of piezocatalysis and thermal (photothermal) catalysis further improved the production rates of •O$_2^-$ and •OH active species, as demonstrated in Fig. 4d–f and Supplementary Figs. 37–39. Thus, synergistic catalytic activity enhancements were obtained. In short, under the stimulation of vibration, heat, or NIR

light, free charges were generated and subsequently reacted with the surface-adsorbed $H_2O$ and $O_2$ molecules to produce $\cdot OH$ and $\cdot O_2^-$ radicals (Fig. 4g). These highly oxidizing species further degraded the organic pollutants to $CO_2$.

Notably, NIR light irradiation and sonication produced almost equal amounts of $\cdot O_2^-$, while sonication displayed much more significant $\cdot OH$ radical production than superoxide radicals. The differences were probably caused by the different reaction mechanisms of the two processes. Under NIR irradiation, hot electrons generated by light heating were captured by oxygen to generate $\cdot O_2^-$, which was further converted into hydrogen peroxide that eventually decomposed into $\cdot OH$ (Supplementary Equations 1–3). This is a multi-step reaction process with a low production efficiency of hydroxyl radicals. In contrast, under sonication, $\cdot OH$ could be directly generated by the reaction between the piezo-generated holes and water. Moreover, $\cdot O_2^-$ was generated by the reaction of oxygen and piezo-generated electrons, which also contributed to the generation of hydroxyl radicals via Supplementary Equations 1–3. As such, sonication produced more $\cdot OH$ than NIR irradiation.

Besides catalytic degradation applications, $Ti_3C_2T_X$ was also analyzed for the piezocatalytic production of $H_2$, which is an ideal energy carrier for realizing a carbon-neutral society. As shown in Fig. 5a, under sonication with a power of 200 W for 3 h, $Ti_3C_2T_X$ showed efficient $H_2$ evolution activity in the presence of a methanol sacrificial agent. The rate of $H_2$ production reached 1341 $\mu mol \cdot g^{-1} \cdot h^{-1}$. The comparison in Fig. 5b is used to highlight the high piezocatalytic efficiency of the $Ti_3C_2T_X$ MXene. These chosen catalysts are widely studied piezoelectric materials with considerable catalytic performances (Supplementary Table 8). The $H_2$ production rate of $Ti_3C_2T_X$ was much higher than those of most reported piezocatalysts for $H_2$ evolution (Fig. 5b). Moreover, almost no $H_2$ was generated over $Ti_3C_2T_X$ without sonication. These comparison results confirmed the $H_2$ production potential over the $Ti_3C_2T_X$ piezocatalyst driven by sonication. In addition, without the sacrificial reagent methanol, $H_2$ production was still observed, but at a much lower reaction rate (135 $\mu mol \cdot g^{-1} \cdot h^{-1}$). The low catalytic activity may have resulted from the rapid recombination of positive and negative carriers in $Ti_3C_2T_X$ without the sacrificial agent[47].

Interestingly, an obvious enhancement in the piezocatalytic $H_2$ evolution activity was observed for $Ti_3C_2T_X$ (Fig. 5c) after several reaction cycles. A similar phenomenon was observed during degradation. As shown in Supplementary Fig. 40, after sonication for 7 h, $Ti_3C_2T_X$ MXene showed improved catalytic activity for MB degradation. To understand this, the morphology of the sample after the reaction was investigated (Fig. 5d and Supplementary Fig. 41). Sonication exfoliated the layered $Ti_3C_2T_X$ MXene (Supplementary Fig. 42) into thinner $Ti_3C_2T_X$ nanosheets, which increased the specific surface area and exposed more active surfaces of the catalyst. Because $H_2$ evolution and MB degradation are heterogeneous catalytic processes occurring at the catalyst surface, a larger surface area with more active sites increased the likelihood of piezo-generated charge carriers interacting with the reactants, thereby enhancing the catalytic performance[9,48,49]. In addition, $H_2$ was produced under mild stirring. As shown in Supplementary Fig. 43, when stirring at 500 rpm at 338 K, an $H_2$ evolution rate of 71 $\mu mol \cdot g^{-1} \cdot h^{-1}$ was obtained. In contrast, most of the traditional piezocatalysts were inactive under such conditions. These results highlight the unique advantage of $Ti_3C_2T_X$ for harvesting and converting abundant mechanical energy.

To further explore the application potential of MXenes, a cost-effectiveness analysis, including the synthesis cost of the material and the energy cost for the catalytic processes, were calculated. For comparison, the costs of some typical piezocatalysts used in Fig. 5b were also evaluated. Supplementary Table 9 and Supplementary Note 6 show that $Ti_3C_2T_X$ had a relatively low synthesis cost of 5.58 ¥/g (¥ refers to Chinese Yuan), which is lower than most previously reported

piezocatalysts, but its catalytic activity was much higher. In addition, Supplementary Table 10 compares the energy costs of $Ti_3C_2T_X$ for various catalytic applications. Among these, the piezocatalytic reaction using hydraulic energy was the most cost-effective process. Nevertheless, the catalytic efficiency was unsatisfactory for large-scale applications, suggesting that future efforts should focus on improving the catalytic activity of the MXene catalysts.

Collectively, the above study demonstrates that $Ti_3C_2T_X$ can use diverse renewable energy sources. Considering the large family of MXene with similar structures, the harvesting and conversion of multiple mechanical, thermal, and solar energy over other MXene for catalytic applications can be expected. To verify this deduction, three other common MXene materials, $V_2CT_X$, $Nb_2CT_X$, and $Ti_2CT_X$, were fabricated via the chemical etching of the same $M_2AlC$ precursor and tested for MB degradation (Supplementary Fig. 44 and Supplementary Note 7). As shown in Fig. 5e, under sonication, all MXene materials showed catalytic degradation activity. $V_2CT_X$ presented the best performance with a degradation rate of 0.078 $min^{-1}$. Moreover, $V_2CT_X$ was further used to degrade MB by utilizing other different energy sources. As shown in Fig. 5f and Supplementary Figs. 45, 46, $V_2CT_X$ efficiently degraded MB under stimulation by heating, stirring, and NIR light. Also, synergistic catalytic activity enhancement of sonication-heating, stirring-heating, and stirring-photothermal was realized over $V_2CT_X$. Furthermore, to examine whether these MXene materials could be used for clean fuel production, the piezocatalytic $H_2$ evolution performance over $V_2CT_X$, $Nb_2CT_X$, and $Ti_2CT_X$ was tested with the assistance of methanol as a sacrificial agent. As shown in Fig. 5g, all MXene materials showed efficient $H_2$ production activities. $V_2CT_X$ presented the highest $H_2$ production rate of 2119 $\mu mol \cdot g^{-1} \cdot h^{-1}$, which was much higher than the performance of $Ti_3C_2T_X$. These results validate the fascinating prospect of utilizing diverse MXene materials for harvesting and converting multiple renewable energy sources.

Here, we demonstrated the simultaneous harvesting and conversion of multiple mechanical, thermal, and solar energy sources over 2D polar $Ti_3C_2T_X$ MXene materials prepared by simple HF etching. $Ti_3C_2T_X$ showed remarkable catalytic performance for organic pollutant decomposition and clean fuel production. Especially, the metallic $Ti_3C_2T_X$ conductor was even active at ambient temperature (298 K) without light or mechanical vibration, while traditional polar semiconductors were completely inactive. Synergistic enhanced piezo-thermal and piezo-photothermal catalysis further demonstrated the advantage of MXene for capturing various energy sources. $V_2CT_X$, $Nb_2CT_X$, and $Ti_2CT_X$ also showed similar multi-functionality. Given the large family and rich chemistry of MXene materials, this work provides an avenue to capture multiple renewable energy sources for achieving a sustainable society.

## Methods
### Materials
Titanium aluminum carbide ($Ti_3AlC_2$), hydrofluoric acid (HF), bismuth nitrate ($Bi(NO_3)_3$), sodium tungstate ($Na_2WO_4$), urea ($CO(NH_2)_2$), cadmium chloride ($CdCl_2$), thiourea, ammonium metavanadate ($NH_4VO_3$), potassium bromide (KBr), zinc nitrate ($Zn(NO_3)_2$), sodium hydroxide (NaOH), methanol ($CH_3OH$), and barium titanate ($BaTiO_3$) were purchased from Sinopharm Chemical Regent Co., Ltd. (Shanghai, China). Deionized (DI) water was obtained from local sources. All the materials were used as received without further purification.

### Synthesis
Preparation of the $Ti_3C_2T_X$ MXene. Briefly, 1 g of LiF was dissolved in 20 mL of 6 M HCl solution in a 250 mL Teflon beaker. Then, $Ti_3AlC_2$ (1 g) was slowly added, followed by reacting at 35 °C for 24 h. The resulting product was collected by centrifugation at 3500 rpm and washed with DI water several times until pH > 6. Then, the $Ti_3C_2T_X$ sediment was dried in an oven at 298 K.

Synthesis of $Ti_3C_2T_X$ MXene film. After obtaining the $Ti_3C_2T_X$ samples, the sediment was re-dispersed in DI water and sonicated for 10 min to delaminate the MXene flakes. Most unexfoliated MXene was removed after centrifugation at 1360×g for 1 h. The colloidal supernatant was collected, and the concentration was determined to be ~0.5 mg/mL. The solution was filtered to form a $Ti_3C_2T_x$ film. The preparation of piezocatalysts have been provided in the Supplementary Methods.

## Characterization

A Bruker D8 Advance X-ray diffractometer was used to analyze the crystal structures of the prepared catalysts. A Cary 500 ultraviolet-visible (UV-vis) diffuse reflectance spectrophotometer (DRS) was utilized to investigate the optical properties of the catalysts. A field-emission scanning electron microscope (FESEM; JSM-6700F) and a transmission electron microscope (TEM; JEM-2010, FEI, Tecnai $G^2$ F20 FEG TEM) were used to determine the micromorphology of the as-synthesized samples. X-ray photoelectron spectroscopy (XPS) was performed using a Thermo Scientific ESCA Lab250 spectrometer consisting of monochromatic Al Kα as the X-ray source. All binding energies were calibrated to the C 1 s peak of surface adventitious carbon at 284.6 eV. The surface area of the samples was measured by the Brunauer–Emmett–Teller (BET) method using nitrogen adsorption and desorption isotherms on a Micrometrics ASAP 2020 system. PFM analysis was performed on a commercial piezoresponse force microscope (Oxford Instruments, MFP-3D). The photothermal effect of the sample was recorded by a Fotric IR thermal imager. Theoretical calculation and additional characterization have been provided in the Supplementary Methods.

## Catalytic activities

**Piezo, thermal, and piezo-thermal catalytic degradation of MB.**
Methylene blue (MB) solution (50 ml, 20 ppm) was added to a 100 mL beaker. Then, 50 mg of catalyst was dispersed in the above solution. The suspension was placed in the dark for 30 min at room temperature (ca. 288 K) without stirring to establish adsorption-desorption equilibrium. For piezocatalysis, the reaction system was sonicated in the dark using an ultrasonic cleaner (KQ-200VDE-DZ, 200 W) equipped with a thermostatic water bath or stirred in the dark (1000 rpm) at 298 K. For thermal catalysis, the suspension was treated in an oil bath at various temperatures without stirring in the dark. For piezo-thermal catalytic process, the suspension was treated at various temperatures with magnetic stirring (1000 rpm) or sonication. During these reactions, 2 mL of reactant suspensions was extracted at certain time intervals and analyzed using UV-vis absorption spectroscopy (UV-5500 PC).

**Photothermal and piezo-photothermal catalytic degradation of MB.**
For the photothermal catalytic reaction, 30 mL of MB solution was added into a 100 mL beaker. A 100 W NIR lamp was used as the light source. The MXene film was submerged in the MB solution in the dark without stirring for 30 min to establish adsorption-desorption equilibrium. After that, the reaction system was irradiated by NIR light with a filter (700–1200 nm) from the top without stirring. For the piezo-photothermal catalytic degradation of MB, the reaction was carried out using the same procedure with a constant stirring of 500 rpm.

**Detection of active oxygen species.** The amounts of $\cdot O_2^-$ and $\cdot OH$ generated under stimulation by different energy sources were determined by NBT transformation and terephthalic acid (TA) photoluminescence (TA-PL) experiments, respectively[50,51]. The reaction conditions were the same as those for MB degradation, except that MB was replaced by TA ($5 \times 10^{-4}$ mol/L) or NBT ($2.5 \times 10^{-5}$ mol/L). NBT can react with $\cdot O_2^-$ and displays maximum absorbance at 259 nm. Tracking the reduction of NBT using a UV-5500 PC spectrophotometer can ascertain the amount of generated $\cdot O_2^-$. TA can react with $\cdot OH$ radicals and generate a product with a fluorescence emission maximum at 425 nm. The amount of $\cdot OH$ radicals was determined by measuring the fluorescence intensity at 425 nm. The details for calculating the concentrations of $\cdot O_2^-$ and $\cdot OH$ radicals are shown in the Supporting Information.

**Hydrogen ($H_2$) production.** Typically, 5 mg of $Ti_3C_2T_X$ was dispersed in 10 mL of a DI water/methanol mixture (10 vol% methanol). The aqueous suspension was sealed in a 25 mL borosilicate tube and purged by $N_2$ for 15 min to completely remove the air. Then, the reaction system was initiated by sonication and then stirred. To detect the amount of generated $H_2$, 1 mL gas was intermittently extracted and analyzed by a gas chromatograph (7890B, Fuli) equipped with a thermal conductivity detector.

## Data availability

The data that support the findings of this study are available from the corresponding author on request. The relevant data generated in this study are provided in the Supplementary Information. Source data are provided with this paper.

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

## Acknowledgements

This work was financially supported by the Natural Science Foundation of China (Grants No. 21905049 and 21802085), Natural Science Foundation of Fujian Province (2019J01735, 2019J01730, and 2020J01201), Program for New Century Excellent Talents in Fujian Province University; the Award Program for Tongjiang Scholar Professorship and the Doctoral Research Start-up Funds Project of Quanzhou Normal University, the Research Foundation of the Academy of Carbon Neutrality of Fujian Normal University (TZH2022-07), and the Award Program for Minjiang Scholar Professorship.

## Author contributions

X.P., M.-Q.Y, S.L., and Z.B. directed the project and wrote the manuscript. X.Y. performed density functional theory calculations. M.Y., X.L., H.K., Q.Q., and X.Z. performed the experiments and analyzed the data. All authors discussed the results and commented on the manuscript.

## Competing interests

The authors declare no competing interests.
