## [Peer Review File · Nature Communications]

Multiple renewable energy harvesting over 2D MXenes polar catalystsREVIEWER COMMENTS

Reviewer #1 (Remarks to the Author):

Authors reported the exploitation of MXene materials as versatile catalysts for multi-energy utilization. The etching of Ti₃AlC₂ shows remarkable catalytic performance for organic pollutants decomposition and H₂ production under the stimulation of diverse energy resources, including photo, thermal and mechanical energy in the forms of vibration, flow, heat, and NIR light. The study reveals that hydroxyl and superoxide radicals can produce on the Ti₃C₂TX under the stimulation of diverse energy sources. However, despite the significant result, some severe problems have not been well proven and explained.

1. The topic has specified "plasmonic polar MXene"; however, "plasmonic" only appears one time in the article "... in which the solar energy is converted to heat through the plasmonic MXene and then thermally activate the MXene." However, the article discussed sonication, thermal excitation, and NIR for dye degradation. If there is a plasmonic in polar MXene, what manipulation of signals at optical frequencies along metal-dielectric interfaces in the nanometer scale of plasmonic MXene? The output performance should be distinguishable for different optical frequencies. However, there is no discussion about these issues. Plasmonic polar MXene is entirely nothing relative to the article
2. In addition, please prove the localized surface plasmon modes supported by polar Mxene (referred to as plasmonics modes).
3. The authors reported that "the relative amplitude image in Figure 2g with the butterfly amplitude loop (Figure 2h) confirms the apparent piezoelectricity in Ti₃C₂TX...". I think the butterfly amplitude loop is very weak, and the amplitude is too small to prove the ferroelectric properties of MXene.
4. The noncentrosymmetric structure should be identified if Ti₃C₂TX belongs to ferroelectric or piezoelectric materials. What is the axis of the net polarization? Figures 1e and 1f are insufficient to prove it.
5. The dye adsorption effect should be investigated to prove there is no dye being adsorption on the surface of the Mxene. The charge surface in the material with dye should be considered. Also, different dyes and electric charges should be investigated.
6. In Figure 4d (left two images), why do the NIR and sonication conditions produce almost equal amounts of superoxides? However, sonication displays a much more significant OH radical than superoxide. The explanation in Figure 4d is insufficient.
7. Utilized the NIR light needs to distinguish the light irradiation and thermal issues. The absorption edge for all materials should be measured to understand NIR spectrum is nicely matched with the materials' band gap.
8. In Figure 5f, the authors should explain why the rate constant for the "sonication+338" is more significant than sonication only. But the rate constant for the "NIR+338" is less significant than the sole NIR condition. The thermal excitation effect results are inconsistent between sonication+338 and NIR+338.
9. In section 3.4, the author stated, "The thermal catalytic activity can be attributed to the metallic property of the Ti₃C₂TX. It can activate molecular oxygen to generate reactive oxygen species of •O₂⁻, which can be further converted into •OH via the formation of H₂O₂ intermediate." Please explain in

detail how the molecular oxygen is transformed into $\bullet\text{O}_2^-$. Namely, how can $\text{Ti}_3\text{C}_2\text{Tx}$ MXene undergo thermal catalytic effect to proceed with the charges involved $\bullet\text{O}_2^-$. Reaction mechanisms should be provided.

10. In Figure 5c, the author discussed that the enhanced HER ability could be attributed to more active surfaces exposed during the process. Can the same phenomenon be observed in the degradation process? Namely, will the degradation ability be enhanced after several cycles? If so, what is the primary reason for degradation?

11. In section 3.1, the author stated, "The metallic nature enables the MXene to exhibit catalytic properties similar to the noble metals, which has been verified as efficient thermal catalysts for dehydrogenation and oxidation reactions." References 27 to 29 are not related to this statement. Please provide proper references.

12. In Figure 1a-1d, the calculated band structures show metallic phases with no bandgap. The electrons emitted from the valence band to the conduction band undergo redox reaction for a photoelectric material. As shown in the calculation results, how can the charge separation occur for a material without a bandgap?

13. The AFM image in Figure 2f and the thickness profile in Figure S4b depicts the height of $\text{Ti}_3\text{C}_2\text{Tx}$ MXene. However, the corresponding PFM image demonstrates a relatively low voltage output, and the signal is only apparent on the lower right part of the material. It is hard to prove the piezo-voltage output of a single $\text{Ti}_3\text{C}_2\text{Tx}$ flake.

14. In Figure 3a, though Ti_3AlC_2 only exhibited a weak degradation ability of MB dye, the MB dye removal ratio did increase steadily with time. Can the author provide reasons for this phenomenon?

15. Authors state, "The degradation performance at the initial and later stages is different because the few-layered have been exfoliated. The simulation models in Figure 1 were only used to calculate the monolayer $\text{Ti}_3\text{C}_2\text{Tx}$. Based on the simulation, can the piezoelectric characteristic remain in the thicker $\text{Ti}_3\text{C}_2\text{Tx}$ MXene? Will the monolayer's piezoelectric is higher than the multilayer?"

16. Will the pre-adsorption ability vary under different temperatures? under 298K, 318K, and 338K.

17. The pre-adsorption test was conducted without stirring in the dark. Due to the density of the powder, $\text{Ti}_3\text{C}_2\text{Tx}$ MXene may settle down in the container. This might cause incomplete adsorption of the dyes. Can the author perform the pre-adsorption test under very low-speed magnetic stirring? The lower stirring speed can probably complete the pre-adsorption of $\text{Ti}_3\text{C}_2\text{Tx}$ toward organic dyes without triggering piezoelectric degradation.

18. Both piezo-degradation under sonication and the thermal-degradation ratio of $\text{Ti}_3\text{C}_2\text{Tx}$ can reach more than 90% in Figures 3a and 3d. However, the fluorescence spectra intensities of TA solution over $\text{Ti}_3\text{C}_2\text{Tx}$ in Figure S25a and Figure 27a have a significant difference. Can the author provide the reasons?

19. Based on the previous question, in Figure S27a, the baseline at 0 minutes is not flat, and the whole spectra seem to be asymmetric. This result is quite different from that of Figure S25a. Can the author provide explanations?

Reviewer #2 (Remarks to the Author):

This work is on MXene as catalysts for renewable energy.

Some comments:

- 1) Catalysts are highly specific in most practical applications. Instead of trying to create a material for various functions, it would better to focus on improving the performance for one specific function. Need to better justify the need for a catalyst to satisfy so many functions.
- 2) The writing needs to be significantly improved. Many awkward sentences throughout.
- 3) MXene is a topic of very active research. For comparisons like in Fig 5b, it is necessary to justify why these catalysts were specially chosen for comparison and why others left out. The conditions for the tests also should be clarified for a fair comparison.
- 4) Need to comment on cost-effectiveness for each catalysis function.
- 5) Error bars needed for the figures containing data in fig 3, 4, 5 , etc.
- 6) The longer term stability and reusability need to be commented on since the authors are proposing this for sustainability.

Reply to the comments of reviewers

Reviewer #1:

Authors reported the exploitation of MXene materials as versatile catalysts for multi-energy utilization. The etching of Ti_3AlC_2 shows remarkable catalytic performance for organic pollutants decomposition and H_2 production under the stimulation of diverse energy resources, including photo, thermal and mechanical energy in the forms of vibration, flow, heat, and NIR light. The study reveals that hydroxyl and superoxide radicals can produce on the $\text{Ti}_3\text{C}_2\text{T}_x$ under the stimulation of diverse energy sources. However, despite the significant result, some severe problems have not been well proven and explained.

Comment 1. The topic has specified "plasmonic polar MXene"; however, "plasmonic" only appears one time in the article "... in which the solar energy is converted to heat through the plasmonic MXene and then thermally activate the MXene." However, the article discussed sonication, thermal excitation, and NIR for dye degradation. If there is a plasmonic in polar MXene, what manipulation of signals at optical frequencies along metal-dielectric interfaces in the nanometer scale of plasmonic MXene? The output performance should be distinguishable for different optical frequencies. However, there is no discussion about these issues. Plasmonic polar MXene is entirely nothing relative to the article.

Reply: Thanks for the valuable comment. Previous studies have well proven that MXene exhibits strong localized surface plasmon resonances. The plasmon energy is determined by an interplay between interband transitions and boundary effects correlated with the size and thickness of $\text{Ti}_3\text{C}_2\text{T}_x$ flakes, which offers a potential to tune the plasma frequencies over a large spectral range from the visible to near infrared range (*Phys. Rev. B* 2014, 89, 235428; *ACS Nano* 2018, 12, 8, 8485-8493; *Light Sci. Appl.* 2022, 11, 22).

In order to prove that the catalytic activity in our system is caused by the photothermal conversion of SPR over the $\text{Ti}_3\text{C}_2\text{T}_x$, wavelength-dependent experiments have been carried out. As shown in **Supplementary Fig. 29-31**, it is found that the catalytic activity of the film changes with different wavelength of light, which is consistent with the light absorption spectrum. Importantly, the catalytic activity trends are in accordance with the temperature change caused by the different wavelengths of light irradiation. Moreover, when controlling the reaction temperature at 298 K using a circulating cooling bath to eliminate light heating effect, the activity of the sample under NIR light irradiation is almost the same as the activity tested at 298 K without light irradiation, further verifying that the NIR light-induced activity of the $\text{Ti}_3\text{C}_2\text{T}_x$ MXene is caused by SPR photothermal conversion effect.

Furthermore, we agree with the reviewer that the discussion on plasmonic effect of $\text{Ti}_3\text{C}_2\text{T}_x$ is limited. This is due to that the main research objective of the present work

is the exploitation of MXene materials as versatile catalysts for multi-energy utilization and discloses the underlying working mechanisms. The SPR effect of the Ti_3C_2Tx is not the primary research scope of the paper. In this context, we delete the word of “plasmonic” from the title.

Corresponding Revision:

Title:

Multiple renewable energy harvesting over 2D MXenes polar catalysts

(Page 15, paragraph 2; Supplementary Fig. 29, Fig. 30 and Fig. 31 have been added)

Supplementary Fig. 29 a, b Photothermal catalytic degradation of MB under different wavelengths of light irradiation over Ti_3C_2Tx film.

Supplementary Fig. 30 a Absorption spectrum and action spectrum of MB degradation of Ti_3C_2Tx film. **b** Temperature of Ti_3C_2Tx film under different wavelength of light irradiation.

Supplementary Fig. 31 Catalytic degradation of MB under different conditions over $\text{Ti}_3\text{C}_2\text{T}_x$ film.

Page 15 Paragraph 2:

To clarify the origin of the catalytic activity under NIR irradiation, the action spectrum of $\text{Ti}_3\text{C}_2\text{T}_x$ was obtained. As shown in **Supplementary Fig. 29** and **Supplementary Fig. 30a**, $\text{Ti}_3\text{C}_2\text{T}_x$ displayed an action spectrum that resembled the plasmonic absorption spectrum of the sample. Importantly, the catalytic activity trends were in accordance with temperature changes caused by irradiation with different wavelengths (**Supplementary Fig. 30b**). Moreover, when controlling the reaction temperature at 298 K using a circulating cooling bath to eliminate the effect of light heating, the activity of $\text{Ti}_3\text{C}_2\text{T}_x$ under NIR light irradiation remained almost the same as its activity at 298 K without light irradiation (**Supplementary Fig. 31**). These results verify that the NIR light-induced activity of the $\text{Ti}_3\text{C}_2\text{T}_x$ MXene was caused by SPR photothermal conversion.

Comment 2. In addition, please prove the localized surface plasmon modes supported by polar MXene (referred to as plasmonics modes).

Reply: Thanks for the comment. According to the previous report, the plasmon mode of a nanomaterial can be defined as dipole, quadrupole, or high-order modes (Plasmonics 2016, 11, 37–44). As for MXene, multiple plasmon resonance modes including dipole and multipolar can be found in a wide range of resonance wavelengths, which is closely related to the shape, size, and thickness of the $\text{Ti}_3\text{C}_2\text{T}_x$, as demonstrated by Alshareef et al. (*ACS Nano* 2018, 12, 8485-8493).

In addition, as replied in Comment 1, the main research objective of the present work is the exploitation of MXene materials as versatile catalysts for multi-energy utilization and discloses the underlying working mechanisms. The plasmonic effect of

the $\text{Ti}_3\text{C}_2\text{T}_x$ is not the primary research scope of the paper. Thus, we did not discuss much about the SPR of $\text{Ti}_3\text{C}_2\text{T}_x$. The word of “plasmonic” in the title is also deleted.

Corresponding Revision:

Title:

Multiple renewable energy harvesting over 2D MXenes polar catalysts

(Page 10, paragraph 1)

Literature reports have shown that the plasmon energy is determined by an interplay between interband transitions and boundary effects correlated with the size and thickness of $\text{Ti}_3\text{C}_2\text{T}_x$ flakes. This offers a potential method to tune the plasma frequencies over a large spectral range from the visible to near-infrared region.^{38, 39, 40} Multiple plasmon resonance modes, including dipole and multipolar, were observed over a wide range of resonance wavelengths in $\text{Ti}_3\text{C}_2\text{T}_x$ flakes.⁴⁰

Comment 3. The authors reported that "the relative amplitude image in Figure 2g with the butterfly amplitude loop (Figure 2h) confirms the apparent piezoelectricity in $\text{Ti}_3\text{C}_2\text{T}_x$...". I think the butterfly amplitude loop is very weak, and the amplitude is too small to prove the ferroelectric properties of MXene.

Reply: Thanks. We have re-performed the PFM analysis by modifying the sample preparation method according to the previous report. (*Nano Energy* 2022, 101 107545). As shown in **Figure R1**, the $\text{Ti}_3\text{C}_2\text{T}_x$ displays strong butterfly amplitude loop, clearly proving the ferroelectric properties of the material. The result is also in consistent with the previous investigations on the piezoelectric property of MXene (*Nano Energy* 2021, 90, 106528; *Nanoscale* 2020, 12, 21291-21298).

Figure R1. Amplitude-voltage curve of $\text{Ti}_3\text{C}_2\text{T}_x$.

Corresponding Revision:

(Figure 2 has been revised.)

Figure 2. **f** AFM image, **g** PFM image, and **h** amplitude-voltage curve of $\text{Ti}_3\text{C}_2\text{T}_x$.

Comment 4. The noncentrosymmetric structure should be identified if $\text{Ti}_3\text{C}_2\text{T}_x$ belongs to ferroelectric or piezoelectric materials. What is the axis of the net polarization? Figures 1e and 1f are insufficient to prove it.

Reply: Thanks for the valuable comment. The structure of monolayer Ti_3C_2 MXene belongs to $\text{P}\bar{3}\text{m}1$ space group. This structure exhibits inversion symmetry, which is intrinsically non-piezoelectric. However, when the surface is functionalized by T_x groups ($\text{T} = \text{F}, \text{OH}, \text{O}$), the Wyckoff positions of $\text{Ti}_3\text{C}_2\text{T}_x$ are changed. The space group is moved from $\text{P}\bar{3}\text{m}1$ to $\text{P}3\text{m}1$ (**Supplementary Table 3**). The $\text{P}3\text{m}1$ space group is noncentrosymmetric and therefore enables the $\text{Ti}_3\text{C}_2\text{T}_x$ to display piezoelectric property. The crystal structure of $\text{Ti}_3\text{C}_2\text{T}_x$ lack inversion centers in and out of the plane in the direction of the x - y plane, which causes the $\text{Ti}_3\text{C}_2\text{T}_x$ MXene to produce in-plane piezoelectric properties. The net polarization is along the x - y plane. This has also been verified by previous report of Song et al., as shown in **Figure R2** (*Nano Energy* 2021, 90, 106528).

Supplementary Table 3. The symmetry group of Ti_3C_2 and $\text{Ti}_3\text{C}_2\text{T}_x$ monolayers.

Sample	Space group	Symmetry
Ti_3C_2	$\text{P}\bar{3}\text{m}1$	centrosymmetric
$\text{Ti}_3\text{C}_2\text{F}_2$	$\text{P}3\text{m}1$	Non-centrosymmetric
$\text{Ti}_3\text{C}_2(\text{OH})_2$	$\text{P}3\text{m}1$	Non-centrosymmetric
$\text{Ti}_3\text{C}_2\text{OHF-1}$	$\text{P}3\text{m}1$	Non-centrosymmetric
$\text{Ti}_3\text{C}_2\text{OHF-2}$	$\text{P}3\text{m}1$	Non-centrosymmetric
$\text{Ti}_3\text{C}_2\text{OHF-3}$	$\text{P}3\text{m}1$	Non-centrosymmetric

Figure R2. The testaments of the parallel and series circuits for the monolayer Ti₃C₂T_x MXene devices. a The schematic diagram of the parallel Ti₃C₂T_x MXene devices with the same lattice direction. b The superposition state of the piezoelectric current of the parallel MXene devices with the same lattice direction. c The schematic diagram of the series MXene devices with the same lattice direction. d The superposition state of the piezoelectric response of the series MXene devices with the same lattice direction. e The schematic diagram of the parallel MXene devices with the opposite lattice direction. f The difference state of piezoelectric responses of the

parallel MXene devices with the opposite lattice direction. g The schematic diagram of the series MXene devices with the opposite lattice direction. h The piezoelectric responses of the series MXene devices with the opposite lattice direction (*Nano Energy* 2021, 90, 106528).

Corresponding Revision:

(Page 6, paragraph 2; Supplementary Table 3 have been added)

The transformation from non-piezoelectric Ti_3C_2 to piezoelectric $Ti_3C_2T_X$ originated from changes in the crystal structure. Based on the structure models, monolayer Ti_3C_2 without functional group belongs to the $P\bar{3}m1$ space group, which exhibits an inverted symmetry and is intrinsically non-piezoelectric. As the surface was functionalized by T_X groups ($T = F, OH, O$), the Wyckoff positions of the $Ti_3C_2T_X$ materials changed. The space group of the MXene changed from $P\bar{3}m1$ to $P3m1$, which is noncentrosymmetric, therefore giving $Ti_3C_2T_X$ piezoelectric properties (Supplementary Table 3).

Supplementary Table 3. The symmetry group of Ti_3C_2 and $Ti_3C_2T_X$ monolayers.

Sample	Space group	Symmetry
Ti_3C_2	$P\bar{3}m1$	centrosymmetric
$Ti_3C_2F_2$	$P3m1$	Non-centrosymmetric
$Ti_3C_2(OH)_2$	$P3m1$	Non-centrosymmetric
Ti_3C_2OHF-1	$P3m1$	Non-centrosymmetric
Ti_3C_2OHF-2	$P3m1$	Non-centrosymmetric
Ti_3C_2OHF-3	$P3m1$	Non-centrosymmetric
Ti_3C_2OHF-4	$P3m1$	Non-centrosymmetric

(Page 7, paragraph 1)

Because the $Ti-T_X$ functional groups were located at non-equivalent positions, the crystal structure obviously lacks the inverted symmetry center along the $x-y$ plane, which gave $Ti_3C_2T_X$ MXene in-plane piezoelectric properties, i.e., the generation of a net polarization along the $x-y$ plane.

Comment 5. The dye adsorption effect should be investigated to prove there is no dye being adsorption on the surface of the MXene. The charge surface in the material with dye should be considered. Also, different dyes and electric charges should be investigated.

Reply: Thanks for the comment. The infrared (IR) spectra of MXene before and after the degradation test have been measured to study the dye adsorption on the surface of the catalyst. As shown in **Supplementary Fig. 7**, the $\text{Ti}_3\text{C}_2\text{T}_x$ shows almost identical IR spectra before and after the catalytic reaction. No typical band of MB is observed, indicating that the dye is degraded rather than adsorbed on the surface of MXene.

In addition, the zeta potential of the $\text{Ti}_3\text{C}_2\text{T}_x$ MXene is measured, which reveals a negatively charged surface (**Supplementary Fig. 19a**). Moreover, the adsorption of different dyes over the MXene is tested. As shown in **Supplementary Fig. 19b**, the $\text{Ti}_3\text{C}_2\text{T}_x$ displays a relatively strong adsorption of positively charged MB due to the electrostatic attraction between them. Nevertheless, the adsorption amount is still less than 8%. This might be caused by the low specific surface area of the $\text{Ti}_3\text{C}_2\text{T}_x$ ($2.8 \text{ m}^2/\text{g}$). The result further indicates that the removal of the dyes should be caused by catalytic degradation.

Corresponding Revision:

(Page 11, paragraph 1; Supplementary Fig. 7, Fig. 19 have been added)

Supplementary Fig. 7 FTIR spectra of $\text{Ti}_3\text{C}_2\text{T}_x$ before and after reaction.

In addition, the IR spectra of the $\text{Ti}_3\text{C}_2\text{T}_x$ before and after the catalytic reaction were measured to study the dye adsorption on the catalyst surface (**Supplementary Fig. 7**). No typical band for MB was observed. The identical IR spectra indicate that MB was degraded rather than adsorbed on the surface of $\text{Ti}_3\text{C}_2\text{T}_x$.

(Page 13, paragraph 2; Supplementary Fig. 19 have been added)

Supplementary Fig. 19 a Zeta potential of Ti₃C₂T_x; b remaining concentration of dye solution after 1 hour pre-adsorption in the presence of Ti₃C₂T_x.

To confirm this inference, control experiments were performed. The surface charge of the Ti₃C₂T_x was first investigated by zeta potential analysis (**Supplementary Fig. 19a**), which showed that the surface of MXene was negatively charged. Nevertheless, due to the small surface area of Ti₃C₂T_x, the adsorption capacities for three different dyes were low (**Supplementary Fig. 19b**). In addition, the possible influence of mass transfer on the adsorption process was also investigated (**Supplementary Fig. 19c**). The adsorption capacity of MXene without stirring was comparable to that of MXene during low-speed stirring (10 rpm). Therefore, it is suggested that Ti₃C₂T_x can utilize ambient heat to catalyze MB degradation rather than adsorbing MB.

Comment 6. In Figure 4d (left two images), why do the NIR and sonication conditions produce almost equal amounts of superoxides? However, sonication displays a much more significant OH radical than superoxide. The explanation in Figure 4d is insufficient.

Reply: Thanks for the comment. The differences in the generation of •OH and •O₂⁻ by the NIR light irradiation and sonication should be due to the different reaction mechanisms. Under the irradiation of NIR light, hot electrons generated by light heating are captured by oxygen to generate •O₂⁻, which are further converted into hydrogen peroxide that eventually decomposes into •OH (**equation S1-S3**). This is a multi-step reaction process. The production efficiency of •OH is not high.

In contrast, under the stimulation of sonication, the •OH could be directly generated by the reaction between the piezo-generated holes and water. Moreover, the •O₂⁻ are generated by the reaction of oxygen and piezo-generated electrons, which can also contribute to the generation of •OH following the same way of **equation S1-S3**. As such, the sonication condition can produce more •OH than the NIR irradiation.

Corresponding Revision:**(Page 19, Paragraph 2)**

Notably, NIR light irradiation and sonication produced almost equal amounts of $\bullet\text{O}_2^-$, while sonication displayed much more significant $\bullet\text{OH}$ radical production than superoxide radicals. The differences were probably caused by the different reaction mechanisms of the two processes. Under NIR irradiation, hot electrons generated by light heating were captured by oxygen to generate $\bullet\text{O}_2^-$, which was further converted into hydrogen peroxide that eventually decomposed into $\bullet\text{OH}$ (**Equations S1–S3**). This is a multi-step reaction process with a low production efficiency of hydroxyl radicals. In contrast, under sonication, $\bullet\text{OH}$ could be directly generated by the reaction between the piezo-generated holes and water. Moreover, $\bullet\text{O}_2^-$ was generated by the reaction of oxygen and piezo-generated electrons, which also contributed to the generation of hydroxyl radicals via **Equations S1–S3**. As such, sonication produced more $\bullet\text{OH}$ than NIR irradiation.

Comment 7. Utilized the NIR light needs to distinguish the light irradiation and thermal issues. The absorption edge for all materials should be measured to understand NIR spectrum is nicely matched with the materials' band gap.

Reply: Many thanks. According to the suggestion, we have investigated the catalytic activity of the MXene under NIR light at 298 K using a circulating cooling bath to exclude the light heating effect. As can be seen from **Supplementary Fig. 31**, when the reaction temperature is decreased to 298 K, the performance of the $\text{Ti}_3\text{C}_2\text{T}_x$ is significantly decreased. The reactivity is almost the same as that the sample tested under thermal excitation at 298 K. Therefore, the origin of the NIR activity is mainly caused by the light-induced thermal catalytic effect rather than light excitation (i.e., photocatalytic effect).

Moreover, the UV-vis spectra of the other samples are also investigated, as shown in **Supplementary Fig. 27**. The results reveal that all these semiconductors show no NIR light absorption. Consequently, these traditional polar catalysts show no activity under NIR light irradiation. The comparison result highlights the advantage of the $\text{Ti}_3\text{C}_2\text{T}_x$ MXene material in capturing and conversion of NIR light.

Corresponding Revision:**(Page 15 Paragraph 1/2, Supplementary Fig. 27 and 31 have been added.)**

Supplementary Fig. 31 Catalytic degradation of MB under different conditions over $\text{Ti}_3\text{C}_2\text{T}_x$ film.

Supplementary Fig. 27 a UV-vis diffuse reflectance spectra, and b NIR light-driven degradation of MB over the different polar semiconductors.

Page 15 Paragraph 1:

Furthermore, a series of traditional piezocatalysts was prepared for comparison (**Supplementary Fig. 27a**). No NIR activity was detected for these catalysts because they were all incapable of harvesting NIR light (**Supplementary Fig. 27b**).

Page 15 Paragraph 2:

Moreover, when controlling the reaction temperature at 298 K using a circulating cooling bath to eliminate the effect of light heating, the activity of $\text{Ti}_3\text{C}_2\text{T}_x$ under NIR light irradiation remained almost the same as its activity at 298 K without light irradiation (**Supplementary Fig. 31**). These results verify that the NIR light-induced activity of the $\text{Ti}_3\text{C}_2\text{T}_x$ MXene was caused by SPR photothermal conversion.

Comment 8. In Figure 5f, the authors should explain why the rate constant for the "sonication+338" is more significant than sonication only. But the rate constant for the "NIR+338" is less significant than the sole NIR condition. The thermal excitation effect results are inconsistent between sonication+338 and NIR+338.

Reply: Many thanks for the comment. The data of "NIR+338" is wrongly marked. It should be "NIR+Stir". We are very sorry for the mistake. The whole manuscript has been carefully checked to avoid similar mistakes. Moreover, the rate constant of "NIR+Stir" is higher than that of sole NIR irradiation. For your reference, the original and revised **Figure 5f** are listed as below.

Origin Figure 5f:

Revised Figure 5f:

Corresponding Revision:

Revised Figure 5f:

Comment 9. In section 3.4, the author stated, "The thermal catalytic activity can be attributed to the metallic property of the $\text{Ti}_3\text{C}_2\text{T}_x$. It can activate molecular oxygen to generate reactive oxygen species of $\bullet\text{O}_2^-$, which can be further converted into $\bullet\text{OH}$ via the formation of H_2O_2 intermediate." Please explain in detail how the molecular oxygen is transformed into $\bullet\text{O}_2^-$. Namely, how can $\text{Ti}_3\text{C}_2\text{T}_x$ MXene undergo thermal catalytic effect to proceed with the charges involved $\bullet\text{O}_2^-$. Reaction mechanisms should be provided.

Reply: Thanks for the valuable comment. The generation of $\bullet\text{O}_2^-$ has been verified by the radical detection experiment, as shown in **Supplementary Fig. 34b**. To understand the underlying mechanism for the thermal catalytic generation of $\bullet\text{O}_2^-$ over the $\text{Ti}_3\text{C}_2\text{T}_x$ MXene, DFT calculations has been carried out. Previous study shows that the thermal activation of O_2 formation of $\bullet\text{O}_2^-$ radicals can be investigated by calculating the Bader charge difference between the free O_2 and the O_2 adsorbed on the surface of a catalyst. The charge transfer of 0.5 |e| is generally required for the generation of $\bullet\text{O}_2^-$ (*ACS Catal.* 2021, 11, 5974-5983). **Supplementary Fig. 35** and **Supplementary Table 6** show the change of Bader charge between the free O_2 molecule and that in the monolayer and bilayer $\text{Ti}_3\text{C}_2\text{T}_x$. The charge density differences reveal that the electron could transfer from the $\text{Ti}_3\text{C}_2\text{T}_x$ catalyst to the adsorbed O_2 molecule. The charge-transfer quantities are 0.85 and 0.83 |e| for the monolayer and bilayer $\text{Ti}_3\text{C}_2\text{T}_x$, respectively, validating that the $\text{Ti}_3\text{C}_2\text{T}_x$ is able to thermally activate O_2 .

Corresponding Revision:

(Page 18, paragraph 2, Supplementary Fig. 35 and Table 6 have been added)

Supplementary Fig. 35 Configurations and charge density difference of O_2 adsorbed on the monolayer and bilayer $\text{Ti}_3\text{C}_2\text{T}_x$.

Supplementary Table 6. Bader charge transfer and charge density difference with

Ti₃C₂T_X.

Atom	Molecule	Monolayer	Bilayer
O1	5.97	6.54	6.28
O2	6.01	6.28	6.53
Total bader charge difference	—	0.85	0.83
O-O (Å)	1.23	1.31	1.31
O-Ti (Å)	—	1.83	1.82
E-ad (eV)	—	-3.622	-3.557

Page 18, paragraph 2:

To further understand the underlying mechanism of the thermal catalytic process, DFT calculations were carried out. Theoretically, the thermal activation of O₂ to form •O₂⁻ can be investigated by calculating the Bader charge difference between free O₂ and O₂ adsorbed on the surface of a catalyst. Generally, the charge transfer of 0.5 |e| is enough to generate •O₂⁻.⁴³ **Supplementary Fig. 35** and **Supplementary Table 7** show the DFT calculation results of changes in the Bader charge between the free O₂ molecule and that in monolayer and bilayer Ti₃C₂T_X. The charge density differences reveal that electrons may have transferred from Ti₃C₂T_X to adsorbed O₂ molecules. The charge-transfer quantities were 0.85 and 0.83 |e| for monolayer and bilayer Ti₃C₂T_X, respectively, confirming that Ti₃C₂T_X could thermally activate O₂.

Comment 10. In Figure 5c, the author discussed that the enhanced HER ability could be attributed to more active surfaces exposed during the process. Can the same phenomenon be observed in the degradation process? Namely, will the degradation ability be enhanced after several cycles? If so, what is the primary reason for degradation?

Reply: Thanks for the comment. In the H₂ evolution reaction, the activity of the catalyst was obviously improved after 7 hours of ultrasonic treatment. To study whether the same phenomenon occurs in the degradation process, the Ti₃C₂T_X after 7 hours of ultrasonic treatment has been tested for MB degradation. As displayed in **Supplementary Fig. 40**, it can be found that the activity of the catalyst is obviously enhanced. The improved catalytic performance can be attributed to the exfoliation of the Ti₃C₂T_X (**Supplementary Fig. 41**), which increases the specific surface area and exposes more active surfaces of the catalyst. Owing to that the H₂ evolution reaction and the MB degradation are heterogeneous catalytic processes taking place at the surface of a catalyst, the large surface area with more active sites can increase the likelihood of piezo-generated charge carriers interacting with the reactants, thereby enhancing the catalytic performance. Similar phenomenon is widely observed in layered materials. (*Adv. Mater.* 2021, 33, 2101751; *Chem. Soc. Rev.* 2016, 45, 4873-4891; *Angew. Chem. Int. Ed.* 2014, 53, 1-6)

Corresponding Revision:

(Page 21, paragraph 2, Supplementary Fig. 40, 41 have been added)

Supplementary Fig. 40 Piezocatalytic degradation of MB over the $Ti_3C_2T_x$ before and after sonication.

Supplementary Fig. 41 TEM image of $Ti_3C_2T_x$ after sonication.

Page 21, paragraph 2:

Interestingly, an obvious enhancement in the piezocatalytic H_2 evolution activity was observed for $Ti_3C_2T_x$ (Figure 5c) after several reaction cycles. A similar phenomenon was observed during degradation. As shown in Supplementary Fig. 40, after sonication for 7 h, $Ti_3C_2T_x$ MXene showed improved catalytic activity for MB degradation. To understand this, the morphology of the sample after the reaction was

investigated (**Figure 5d** and **Supplementary Fig. 41**). Sonication exfoliated the layered $\text{Ti}_3\text{C}_2\text{T}_x$ MXene (**Supplementary Fig. 42**) into thinner $\text{Ti}_3\text{C}_2\text{T}_x$ nanosheets, which increased the specific surface area and exposed more active surfaces of the catalyst. Because H_2 evolution and MB degradation are heterogeneous catalytic processes occurring at the catalyst surface, a larger surface area with more active sites increased the likelihood of piezo-generated charge carriers interacting with the reactants, thereby enhancing the catalytic performance.^{45, 46, 47}

Comment 11. In section 3.1, the author stated, "The metallic nature enables the MXene to exhibit catalytic properties similar to the noble metals, which has been verified as efficient thermal catalysts for dehydrogenation and oxidation reactions." References 27 to 29 are not related to this statement. Please provide proper references.

Reply: Many thanks for the comment. The references have been revised. In addition, the description has been revised to make it more clear.

Corresponding Revision:

(Page 7, Paragraph 1)

The metallic MXene demonstrated high thermal catalytic dehydrogenation performance comparable to that of noble metals.²⁸

Comment 12. In Figure 1a-1d, the calculated band structures show metallic phases with no bandgap. The electrons emitted from the valence band to the conduction band undergo redox reaction for a photoelectric material. As shown in the calculation results, how can the charge separation occur for a material without a bandgap?

Reply: Thanks for the comment. When external strain acts on the $\text{Ti}_3\text{C}_2\text{T}_x$ MXene, the positive and negative charges polarize and produce a piezoelectric field in the corresponding direction. This electric field could effectively separate the charge carriers, as demonstrated by the previous research of Song, et al. (**Figure R3**, *Nano Energy* 2021, 90, 106528). Such stress induced electric field is commonly observed on piezoelectric materials (ACS Appl. Energy Mater. 2022, 5, 9, 11472–11482; ACS Appl. Nano Mater. 2022, 5, 5, 7588–7597; Environ. Sci. Technol. 2018, 52, 14, 7842–7848).

Figure R3. The band diagrams of the piezoelectric effect in the monolayer $\text{Ti}_3\text{C}_2\text{T}_x$ MXene, indicating the behaviors of the piezoelectric polarization and the changes of Schottky height (*Nano Energy*, 2021, 2021, 106528).

Corresponding Revision:

(Page 17 Paragraph 2)

The positive and negative charges polarize and produce a piezoelectric field in the corresponding direction that separates charge carriers that drive catalytic redox reactions.

Comment 13. The AFM image in Figure 2f and the thickness profile in Supplementary Fig. 4b depicts the height of $\text{Ti}_3\text{C}_2\text{T}_x$ MXene. However, the corresponding PFM image demonstrates a relatively low voltage output, and the signal is only apparent on the lower right part of the material. It is hard to prove the piezo-voltage output of a single $\text{Ti}_3\text{C}_2\text{T}_x$ flake.

Reply: Thanks for the comment. We have re-performed the PFM analysis by modifying the sample preparation method according to the previous report. (*Nano Energy* 2022, 101 107545). As shown in **Figure R1**, the $\text{Ti}_3\text{C}_2\text{T}_x$ displays strong butterfly amplitude loop, clearly proving the ferroelectric properties of the material. The result is also in consistent with the previous investigations on the piezoelectric property of MXene (*Nano Energy* 2021, 90, 106528; *Nanoscale* 2020, 12, 21291-21298).

Figure R1. Amplitude-voltage curve of $\text{Ti}_3\text{C}_2\text{T}_x$.

Corresponding Revision:

(Figure 2 has been revised.)

Figure 2. **f** AFM image, **g** PFM image, and **h** amplitude-voltage curve of $\text{Ti}_3\text{C}_2\text{T}_x$.

Comment 14. In Figure 3a, though Ti_3AlC_2 only exhibited a weak degradation ability of MB dye, the MB dye removal ratio did increase steadily with time. Can the author provide reasons for this phenomenon?

Reply: Many thanks for the comment. The weak activity of the Ti_3AlC_2 might be caused by the leaching out of Al under sonication and the surface functionalization of oxygen-containing functional group. As evidenced by the ICP analysis, noticeable Al^{3+} is detected from the reaction solution of Ti_3AlC_2 after the catalytic activity test (**0.5 mg/L**). In addition, the XPS characterization (**Supplementary Fig. 8.**) of the used Ti_3AlC_2 shows that obvious Ti-O bond is generated on the surface, indicating the partial transformation of Ti_3AlC_2 to $\text{Ti}_3\text{C}_2\text{T}_x$ with O and OH groups during the catalytic reaction. This would be the reason accounting for the weak activity of the Ti_3AlC_2 .

Corresponding Revision:

(Page 11, paragraph 1, Supplementary Fig. 8 has been added.)

Moreover, in the presence of Ti_3AlC_2 , the absorption peak intensity of MB was slightly decreased. The weak activity of Ti_3AlC_2 might be caused by the leaching of Al during sonication and surface functionalization by oxygen-containing functional groups. As evidenced by the ICP analysis, noticeable Al^{3+} was detected in the reaction solution of Ti_3AlC_2 after the catalytic activity test (0.5 mg/L). In addition, the XPS characterization of the used Ti_3AlC_2 showed that obvious Ti-O bonds were generated on the surface (**Supplementary Fig. 8**), indicating the partial transformation of Ti_3AlC_2 to $\text{Ti}_3\text{C}_2\text{T}_x$ with O and OH groups during the catalytic reaction. This was the reason for the weak activity of Ti_3AlC_2 .

Supplementary Fig. 8 XPS spectra of Ti_3AlC_2 after sonication: **a**, survey spectra; **b**, O 1s.

Comment 15. Authors state, "The degradation performance at the initial and later stages is different because the few-layered have been exfoliated. The simulation models in Figure 1 were only used to calculate the monolayer $\text{Ti}_3\text{C}_2\text{T}_x$. Based on the simulation, can the piezoelectric characteristic remain in the thicker $\text{Ti}_3\text{C}_2\text{T}_x$ MXene? Will the monolayer's piezoelectric is higher than the multilayer?"

Reply: Thanks for the valuable comment. We have investigated the dipole moment of the multilayer $\text{Ti}_3\text{C}_2\text{T}_x$ using DFT calculations. As presented in **Supplementary Table 2**, the dipole moment of the multilayer $\text{Ti}_3\text{C}_2\text{T}_x$ (5 layers) is comparable to that of the monolayer $\text{Ti}_3\text{C}_2\text{T}_x$.

Corresponding Revision:

(Page 18, paragraph 2, **Supplementary Table 2** have been added)

We have also investigated the influence of the number of layers on the dipole moment of $\text{Ti}_3\text{C}_2\text{T}_x$. The dipole moment of multilayer $\text{Ti}_3\text{C}_2\text{T}_x$ (5 layers) was comparable to that of monolayer $\text{Ti}_3\text{C}_2\text{T}_x$, suggesting that the number of layers had little effect on the dipole moment (**Supplementary Table 2**).

Supplementary Table 2. Calculated dipole moments of and $\text{Ti}_3\text{C}_2\text{OHF-1}$ monolayers and $\text{Ti}_3\text{C}_2\text{OHF-1}$ multilayers (5 layers).

Dipole moment	x(eÅ)	y(eÅ)	z(eÅ)
$\text{Ti}_3\text{C}_2\text{OHF-1}$ monolayers	-0.017	0.000	-0.253
$\text{Ti}_3\text{C}_2\text{OHF-1}$ 5 layers	-0.018	0.000	-0.238

Note: the atomic structure of the $\text{Ti}_3\text{C}_2\text{OHF-1}$ is shown in the Supplementary Fig. 1.

Comment 16. Will the pre-adsorption ability vary under different temperatures? under 298K, 318K, and 338K.

Reply: Thanks. We have tried to test the pre-adsorption of MB dye over the $\text{Ti}_3\text{C}_2\text{T}_x$ at 298K, 318K, and 338K. However, owing to that the dye degradation is immediately occurred under heating condition in the presence of $\text{Ti}_3\text{C}_2\text{T}_x$, the increasing of reaction temperature accelerates the degradation process. In such case, it is hard to evaluate the pre-adsorption ability of the $\text{Ti}_3\text{C}_2\text{T}_x$ under different temperatures. Instead, the pre-adsorption experiments of the catalyst for thermocatalytic reaction are all performed at room temperature.

Comment 17. The pre-adsorption test was conducted without stirring in the dark. Due to the density of the powder, $\text{Ti}_3\text{C}_2\text{T}_x$ MXene may settle down in the container. This might cause incomplete adsorption of the dyes. Can the author perform the pre-adsorption test under very low-speed magnetic stirring? The lower stirring speed can probably complete the pre-adsorption of $\text{Ti}_3\text{C}_2\text{T}_x$ toward organic dyes without triggering piezoelectric degradation.

Reply: Thanks for the comment. According to the suggestion, the pre-adsorption of MB over the $\text{Ti}_3\text{C}_2\text{T}_x$ under a low stirring speed of 50 rpm has been carried out at room temperature (ca. 288 K). As shown in **Supplementary Fig. 19c**, the adsorption capacity of MXene is comparable to that without stirring (0 rpm). Thus, the condition of without stirring in the dark is applicable to complete the pre-adsorption of organic dyes over the $\text{Ti}_3\text{C}_2\text{T}_x$.

Corresponding Revision:

(Page 12, Paragraph 2, Supplementary Fig. 19c has been added.)

Supplementary Fig. 19 c the adsorption curves for MB over the $\text{Ti}_3\text{C}_2\text{T}_x$ with or without stirring.

In addition, the possible influence of mass transfer on the adsorption process was also investigated (**Supplementary Fig. 19c**). The adsorption capacity of MXene without stirring was comparable to that of MXene during low-speed stirring (50 rpm). Therefore, it is suggested that $Ti_3C_2T_x$ can utilize ambient heat to catalyze MB degradation rather than adsorbing MB.

Comment 18. Both piezo-degradation under sonication and the thermal-degradation ratio of $Ti_3C_2T_x$ can reach more than 90% in Figures 3a and 3d. However, the fluorescence spectra intensities of TA solution over $Ti_3C_2T_x$ in Supplementary Fig. 25a and Figure 27a have a significant difference. Can the author provide the reasons?

Reply: Thanks for the comment. The difference in the fluorescence spectra intensities of TA solution can be attributed to the different generation mechanisms of $\bullet OH$ in the sonication reaction and the thermal catalytic process. Specifically, under the stimulation of sonication, the $\bullet OH$ could be directly generated by the reaction between the piezo-generated holes and water. In addition, the $\bullet O_2^-$ generated by the reaction of O_2 and piezo-generated electrons can also contribute to the generation of $\bullet OH$ following the **equation S1-S3**. Consequently, the sonication condition can produce abundant $\bullet OH$.

In contrast, under heating condition, the $\bullet OH$ is only generated from the decomposition of $\bullet O_2^-$ produced by the thermal activation of O_2 over the $Ti_3C_2T_x$ (**equation S1-S3**). This is a multi-step reaction process. The production efficiency of $\bullet OH$ is low. Nevertheless, for the thermal catalytic degradation of MB, the increase of reaction temperature can accelerate the reaction rate according to the Arrhenius equation: $k=Ae^{(-E_a/RT)}$ (k is the rate constant, E_a is the activation energy for the reaction, T is the reaction temperature). As a result, although the production efficiency of $\bullet OH$ is lower, the thermal catalytic process can still display comparable MB degradation activity as that of the sonication reaction.

Corresponding Revision:

(Page 18 Paragraph 2)

Notably, the concentration of free radicals ($\bullet O_2^-$ and $\bullet OH$ radicals) produced by $Ti_3C_2T_x$ under heating at 338 K was lower than that obtained by sonication at 298 K. The two catalytic processes showed similar activities for MB degradation because increasing the reaction temperature facilitated the production of oxygen radicals and

also accelerated the reaction according to the Arrhenius formula: $k = Ae^{-E_a/RT}$ (where, k is the rate constant, E_a is the activation energy, and T is the reaction temperature).

Comment 19. Based on the previous question, in FigureS 27a, the baseline at 0 minutes is not flat, and the whole spectra seem to be asymmetric. This result is quite different from that of Figure S25a. Can the author provide explanations?

Reply: Thanks for the comment. This phenomenon is due to the influence of the photoluminescence of terephthalic acid (TA). Before reaction, TA is excited under 315 nm to produce a emission peak at ca. 350 nm (*J. Mater. Chem. A* 2016, 4, 869-876). Therefore, the baseline at 0 minute is not flat. After a period of reaction, TA reacts with $\bullet\text{OH}$ to produce hydroxyl terephthalic acid (HTA) with a PL peak at ca. 420 nm. When the peak intensity of the HTA is weak, the TA signal is noticeable. The interaction between the two PL peaks leads to the asymmetry of the HTA peak at 420 nm. However, when the intensity of the HTA is very strong (**Supplementary Fig. 25a**), the influence of the TA peak at 350 nm will be greatly weakened, thus resulting in the symmetry of the HTA peak. Considering that the fluorescence peak at 350 nm is not the emission signal of $\bullet\text{OH}$, we do not show this signal in the **Figures S25** and **S27**. For your reference, we re-drawn the **Supplementary Fig. 27a** as shown below.

Supplementary Fig. 34 a Fluorescence spectra of TA solution over $\text{Ti}_3\text{C}_2\text{T}_x$ without stirring in the dark at 338 K.

Corresponding Revision:

Supplementary Fig. 34 a Fluorescence spectra of TA solution over $\text{Ti}_3\text{C}_2\text{T}_x$ without stirring in the dark at 338 K. **b** The absorbance of NBT solution over $\text{Ti}_3\text{C}_2\text{T}_x$ without stirring in the dark at 338 K.

Note: Before reaction, TA is excited under 315 nm to produce a emission peak at ca. 350 nm. Therefore, the baseline at 0 minute is not flat (**Supplementary Fig. 34a**). After a period of reaction, TA reacts with $\bullet\text{OH}$ to produce hydroxyl terephthalic acid (HTA) with a PL peak at ca. 420 nm. When the peak intensity of the HTA is weak, the TA signal is noticeable. The interaction between the two PL peaks leads to the asymmetry of the HTA peak at 420 nm. However, when the intensity of the HTA is very strong (**Supplementary Fig. 32a**), the influence of the TA peak at 350 nm will be greatly weakened, thus resulting in the symmetry of the HTA peak.

Reviewer #2 (Remarks to the Author):

This work is on MXene as catalysts for renewable energy.

Some comments:

Comment 1. Catalysts are highly specific in most practical applications. Instead of trying to create a material for various functions, it would better to focus on improving the performance for one specific function. Need to better justify the need for a catalyst to satisfy so many functions.

Reply: Many thanks. In this manuscript, MXene is used for two kinds of catalytic reactions: dye degradation and H₂ production. Different kind of energy sources (light, heat and vibration) have been utilized to drive the catalytic reaction.

In fact, these energy sources are usually coexist in practical applications. For example, during the photocatalytic reaction, sunlight irradiation is associated with obviously thermal effect. At the same time, in order to accelerate mass transfer in the reaction process, the reaction system would be stirred by a magnetic stirrer, which provides vibration energy for the catalyst. If all these energy sources can be fully utilized, the catalytic activity will be effectively improved.

The main novelty of the present work is the exploitation of MXene materials as versatile catalysts for multi-energy utilization and the disclosing of the underlying working mechanisms.

The motivation for the research has been justified in the introduction of the paper. For your reference, the main contents are listed as below:

The rapid depletion of fossil fuels along with the worsening environment spurs continuous research endeavors in the development of diverse renewable energy harvesting technologies such as piezo, photo, and thermal catalysis to convert intermittent mechanical, solar, and thermal energy storable chemical energy. Unfortunately, due to the unpredictable availability of single-source renewable energy that depends on season, climate and geographical position, the conversion efficiency and stability provided by these individual energy harvesting technologies are still insufficient for practical deployment. In addition, the conventionally designed energy harvester is normally with good performance for one energy sources utilization, but are poorly effective in capture of other energy resources.

To break through this limitation, a promising strategy is to develop advanced energy harvesters that can efficiently capture multiple energy sources. Construction of hybrid devices by the embedding of individual harvesters made from different materials into one system is the most widely adopted strategy, but it is commonly restricted by various material incompatibility and fabrication challenges. As such, the departure from the traditional paradigm of fabricating hybrid device into the development of single composition energy harvester has been emerged and aroused more and more research attention. Nevertheless, realization of such multifunctionality remains daunting due to the demand of satisfying different energy conversion

principles simultaneously within a single material. Especially, the different energy conversion effects should be independent or coupled, instead of counteracting each other.

In this context, we herein for the first time theoretically and experimentally validate the multiple energy source utilization over MXene materials. The underlying mechanisms for the harvesting of mechanical, thermal, and solar energies via piezo, thermal, and photothermal catalysis over the $Ti_3C_2T_X$ has been studied. Moreover, similar multifunctionalities has been validated on Ti_2CT_X , V_2CT_X and Nb_2CT_X MXene materials. Given the large family and rich chemistry of the MXene materials, the progress in this work suggests an avenue to capture multiple renewable energy for approaching the sustainable society.

Comment 2. The writing needs to be significantly improved. Many awkward sentences throughout.

Reply: Many thanks. The language has been polished by native English speaker.

Corresponding Revision:

Please see the marked up revised manuscript for details.

Comment 3. MXene is a topic of very active research. For comparisons like in Fig 5b, it is necessary to justify why these catalysts were specially chosen for comparison and why others left out. The conditions for the tests also should be clarified for a fair comparison.

Reply: Thanks for the comment. The comparison in **Figure 5b** is used to highlight the high piezocatalytic efficiency of the $Ti_3C_2T_X$ MXene. These chosen catalysts are widely studied piezoelectric materials with considerable catalytic performances according to a survey of literature, including MoS_2 (*Adv. Mater.* 2016, 28, 3718-3725), WS_2 , WSe_2 (*Adv. Mater.* 2020, 32, 34), $BiFeO_3$ (*Angew. Chem. Int. Ed.* 2019, 58, 11779), $Pd-BiFeO_3$ (*ACS Appl. Mater. Interfaces* 2021, 13, 15305-15314), $BaTiO_3$ (*Angew. Chem. Int. Ed.* 2021, 60, 16019), $MAPbI_3$ (*Adv. Energy Mater.* 2019, 9, 1901801), ZnO (*Nano Energy* 2019, 62, 376-383), CdS and GaN (*ACS Appl. Mater. Interfaces* 2021, 13, 10916-10924), $Sr_{0.5}Ba_{0.5}Nb_2O_6$ (*ACS Appl. Mater. Interfaces* 2021, 13, 7259-7267)

In addition, the conditions for the tests have been provided in **Supplementary Table 8.**

Corresponding Revision:

(Page 20 Paragraph 1, Supplementary Table 8.)

The comparison in **Figure 5b** is used to highlight the high piezocatalytic efficiency of the $Ti_3C_2T_X$ MXene. These chosen catalysts are widely studied piezoelectric materials

with considerable catalytic performances (**Supplementary Table 8**). The H₂ production rate of Ti₃C₂T_x was much higher than those of most reported piezocatalysts for H₂ evolution (**Figure 5b**).

Supplementary Table 8. Comparison of piezocatalytic H₂ production over different materials.

NO.	Sample	Energy Source	Sample Dosage	H ₂ production Rate (μmol/g/h)	References
1	Ti ₃ C ₂ T _x	40 kHz 200 W	0.005 g Catalyst, 10mL, 10% methanol solution	2682	This work
2	MoS ₂ nanosheets	40 kHz, 110 W	20 mgcat/100 mL H ₂ O (0.01 M FeSO ₄)	29	26
	WS ₂ nanosheets			15	
	WSe ₂ nanosheets			11	
3	Bi ₂ WO ₆ nanoplates	40 kHz, -	20 mg cat/10 mLH ₂ O (20% v/v TEOA)	191	27
4	CH ₃ NH ₃ PbI ₃	70 W, -	50 mg cat/5.5 mL HI solution (57 wt%)	4	28
5	0.7BiFeO ₃ /0.3 BaTiO ₃	40 kHz, 100 W	30 mg cat/50 mLH ₂ O (CH ₃ OH)	1322	29
6	BaTiO ₃	40 kHz, 100 W	10 mg cat/100 mLH ₂ O (TEOA 15v%)	92	30
7	BiFeO ₃	40 kHz, 100 W	10 mg cat/10 mLNa ₂ SO ₃ (0.05 M)	124	31
8	Pd-BiFeO ₃	40 kHz, 100 W	10 mg cat/10 mLNa ₂ SO ₃ (0.05 M)	1140	32
9	TiO ₂ /ZnO nanowires	ultrasonic: 50 W, light:50 W	220 mg cat/150 mLH ₂ O (CH ₃ OH 20v%)	3	33
10	CdS	100 W, 45 kHz, light: 0.88 mW·cm ²	50 mg cat/80 mLH ₂ O (lactic acid 10v%)	284	34
11	Ni/GaN nanowires	110 W, 40 kHz	2 mg cat/100 mLH ₂ O (TEOA 15v%)	88	35
12	Sr _{0.5} Ba _{0.5} Nb ₂ O ₆ -500 °C	110 W, 40 kHz	10 mg cat/100	109	36

mLH₂O (TEOA
15v%)

Comment 4. Need to comment on cost-effectiveness for each catalysis function.

Reply: Thanks for the comment. The cost-effectiveness analysis including the synthesis cost of the material and the energy cost for the catalytic processes are calculated. For comparison, the costs of some typical piezocatalysts used in **Figure 3b** has also been evaluated. **Supplementary Table 9** shows that the Ti₃C₂T_x has a low synthesis cost of 5.58 ¥/g (¥ refers to Chinese Yuan), which is lower than most of the previously reported piezocatalysts, yet its catalytic activity is much higher. In addition, **Supplementary Table 10** shows the comparison of the energy cost of Ti₃C₂T_x for various catalytic applications. Among these, the piezocatalytic reaction using hydraulic energy is the most cost-effective process. Nevertheless, the catalytic efficiency is unsatisfactory for large-scale application. This suggests that future effort should be focused on improving the catalytic activity of the MXene catalysts.

Corresponding Revision:

(Page 21, pragraph 3, Supplementary Table 9 and 10 have been added.)

To further explore the application potential of MXenes, a cost-effectiveness analysis including the synthesis cost of the material and the energy cost for the catalytic processes were calculated. For comparison, the costs of some typical piezocatalysts used in **Figure 5b** were also evaluated. **Supplementary Table 9** shows that Ti₃C₂T_x had a relatively low synthesis cost of 5.58 ¥/g (¥ refers to Chinese Yuan), which is lower than most previously reported piezocatalysts, but its catalytic activity was much higher. In addition, **Supplementary Table 10** compares the energy costs of Ti₃C₂T_x for various catalytic applications. Among these, the piezocatalytic reaction using hydraulic energy was the most cost-effective process. Nevertheless, the catalytic efficiency was unsatisfactory for large-scale applications, suggesting that future efforts should focus on improving the catalytic activity of the MXene catalysts.

Supplementary Table 9. The material cost (C_{material}) and energy cost (C_{energy}) for the preparation of different piezocatalysts.

Sample	Materials (CNY ¥)			Yield	Synthesis	Total cost	Ref.
Ti₃C₂T_x	Ti ₃ AlC ₂	1.00 g	1.80 ¥	0.86 g	308 K 24 h (Heat Stirrer)	5.58 ¥/g	This work
	LiF	1.00 g	1.90 ¥				
	HCl	20.00 mL	0.60 ¥				
	C _{material} : 4.30 ¥				C _{energy} : 0.50 ¥		
MoS₂	NaCl	3.75 g	0.08 ¥	0.16 g	493 K 24 h (Oven)	137.44 ¥/g	27
	NaOH	24.00 g	1.15 ¥				
	SiO ₂	2.4.00 g	0.23 ¥				
	MoO ₃	0.144 g	0.15 ¥				
	NH ₃ ·H ₂ O	30.00 ml	0.84 ¥				
	HCl	30.00 ml	1.02 ¥				
	CH ₄ N ₂ S	0.40 g	0.04 ¥				
C _{material} : 3.51 ¥			C _{energy} : 18.48 ¥				
Bi₂WO₆	Bi(NO ₃) ₃ ·5H ₂ O	1.00 mmol	0.33 ¥	0.349 g	433K 20h (Oven)	46.48 ¥/g	28
	Na ₂ WO ₄	1.00 mmol	0.49 ¥				
	C _{material} : 0.82 ¥				C _{energy} : 15.4 ¥		
MAPbI₃	MAI	1.00 mmol	10.43 ¥	0.62 g	343 K 12h (Oven)	52.03 ¥/g	29
	PbI ₂	1.00 mmol	1.38 ¥				
	C ₃ H ₇ NO	100.00 ml	9.00 ¥				
	C ₆ H ₅ Cl	20.00 ml	2.21 ¥				
C _{material} : 23.02 ¥			C _{energy} : 9.24 ¥				
0.7BiFeO₃/ 0.3BaTiO₃	Bi(NO ₃) ₃ ·5H ₂ O	0.51 g	0.19 ¥	4.28 g	473 K 8h (Oven)	1.55 ¥/g	30
	Fe(NO ₃) ₃ ·5H ₂ O	0.42 g	0.03 ¥				
	HNO ₃	2.00 ml	0.10 ¥				
	KOH	15.00 ml	0.10 ¥				
	Ti(OC ₄ H ₉) ₄	0.15 ml	0.04 ¥				
C _{material} : 0.46 ¥			C _{energy} : 6.16 ¥				
BaTiO₃	NaOH	10.00 mol	19.20 ¥	5.22 g	483 K 24h (Oven)	9.25 ¥/g	31
	TiO ₂	1.88 g	9.14 ¥				
	HCl	0.20 mol	0.54 ¥				
	Ba(OH) ₂ ·8H ₂ O	0.04 mol	0.91 ¥				
C _{material} : 29.79 ¥			C _{energy} : 18.48 ¥				
BiFeO₃	Bi(NO ₃) ₃ ·5H ₂ O	2.43 g	0.92 ¥	1.57 g	453 K 48h (Oven) 773 K 2h (Muffle furnace)	30.99 ¥/g	32
	(CH ₂ OH) ₂	100.00 ml	3.70 ¥				
	FeCl ₃ ·5H ₂ O	1.35 g	0.26 ¥				
	NH ₃ ·H ₂ O	50.00 ml	1.40 ¥				
	NaOH	0.2 mol	1.92 ¥				
	C _{material} : 8.20 ¥				C _{energy} : 40.46 ¥		
Pd-BiFeO₃	BiFeO ₃	0.01 g	0.30 ¥	0.016	423 K 3h	180.12 ¥/g	33

				6 g	(Oven)			
	Pd-NCS	2.4 mmol	0.38 ¥			C _{energy} : 2.31 ¥		
	C _{material} : 0.68 ¥							

Note: The energy cost for material synthesis is calculated based on the rated power for the instrument (water bath heated magnetic stirrer: 30 W; oven: 1100 W; muffle furnace: 2500 W).

Supplementary Table 10. The energy cost for catalytic processes.

Reaction condition	Instrument	Power (W)	Cost (1 hour)
Sonication	Ultrasonic cleaner	200	0.14 ¥
Stir	Stirrer	12	0.0084 ¥
Heating	water bath heated magnetic stirrer	30	0.021 ¥
NIR light	NIR lamp	100	0.070 ¥

Comment 5. Error bars needed for the figures containing data in fig 3, 4, 5, etc.

Reply: Thanks for the comment. The error bars have been added.

Corresponding Revision:

Please see the Figures in Manuscript and supporting information for details.

Comment 6. The longer term stability and reusability need to be commented on since the authors are proposing this for sustainability.

Reply: Thanks for the comment. Long term stability of the catalyst has been investigated. As shown in the revised **Figure 5c**, the catalyst is stable after ten cycles.

Corresponding Revision:

(Figure 5c has been revised.)

Figure 5c. The stability of Ti₃C₂T_x under sonication.

Thanks for all the comments of the reviewers to help us improve the quality of our manuscript!

REVIEWERS' COMMENTS

Reviewer #1 (Remarks to the Author):

The authors have well addressed most of questions.

Another minor comment is: When mentioned about....Diverse renewable energy harvesting technologies, including piezocatalysis....”, suggesting some initially original papers using 2D materials should be cited in the introduction to make the readers understand the historical piezocatalysis and evolution.

Reply to the comment of the reviewer

Reviewer #1:

The authors have well addressed most of questions.

Comment 1. Another minor comment is: When mentioned about...Diverse renewable energy harvesting technologies, including piezocatalysis....”, suggesting some initially original papers using 2D materials should be cited in the introduction to make the readers understand the historical piezocatalysis and evolution.

Reply: Thanks for the comment. Some initially original papers including 5 research articles and 3 review articles related to the 2D piezocatalysts have been cited.

Corresponding Revision:

References.

Research articles

7. Wu JM, Chang WE, Chang YT, Chang C-K. Piezo-Catalytic Effect on the Enhancement of the Ultra-High Degradation Activity in the Dark by Single- and Few-Layers MoS₂ Nanoflowers. *Adv. Mater.* **28**, 3718-3725 (2016).
8. Masimukku S, Hu Y-C, Lin Z-H, Chan S-W, Chou T-M, Wu JM. High efficient degradation of dye molecules by PDMS embedded abundant single-layer tungsten disulfide and their antibacterial performance. *Nano Energy* **46**, 338-346 (2018).
18. Lin Y-T, Lai S-N, Wu JM. Simultaneous Piezoelectrocatalytic Hydrogen-Evolution and Degradation of Water Pollutants by Quartz Microrods@Few-Layered MoS₂ Hierarchical Heterostructures. *Adv. Mater.* **32**, (2020).
40. Feng W, *et al.* Piezopotential-driven simulated electrocatalytic nanosystem of ultrasmall MoC quantum dots encapsulated in ultrathin N-doped graphene vesicles for superhigh H₂ production from pure water. *Nano Energy* **75**, 104990 (2020).
47. You H, *et al.* Harvesting the Vibration Energy of BiFeO₃ Nanosheets for Hydrogen Evolution. *Angew. Chem. Int. Ed.* **58**, 11779-11784 (2019).

Review articles

4. Wang J, *et al.* Energy and environmental catalysis driven by stress and temperature-variation. *J. Mater. Chem. A* **9**, 12400-12432 (2021).
6. Tu S, *et al.* Piezocatalysis and Piezo-Photocatalysis: Catalysts Classification and Modification Strategy, Reaction Mechanism, and Practical Application. *Adv. Funct. Mater.* **30**, 2005158 (2020).
10. Kole AK, Karmakar S, Pramanik A, Kumbhakar P. Transition metal

dichalcogenides nanomaterials based piezocatalytic activity: recent progresses and outlook. *Nanotechnology* **34**, 282001 (2023).